# Learning Data-Efficient and Generalizable Neural Operators via Fundamental Physics Knowledge

**Siying Ma**[1], **Mehrdad M. Zadeh**[1], **Mauricio Soroco**[1]
**Wuyang Chen**[1], **Jiguo Cao**[1*], **Vijay Ganesh**[2*]

[1]Simon Fraser University, [2]Georgia Institute of Technology

## Abstract

Recent advances in scientific machine learning (SciML) have enabled neural operators (NOs) to serve as powerful surrogates for modeling the dynamic evolution of physical systems governed by partial differential equations (PDEs). While existing approaches focus primarily on learning simulations from the target PDE, they often overlook more **fundamental physical principles** underlying these equations. Inspired by how numerical solvers are compatible with simulations of different settings of PDEs, we propose a multiphysics training framework that jointly learns from both the original PDEs and their simplified basic forms. Our framework enhances data efficiency, reduces predictive errors, and improves out-of-distribution (OOD) generalization, particularly in scenarios involving shifts of physical parameters and synthetic-to-real transfer. Our method is architecture-agnostic and demonstrates **consistent improvements** in normalized root mean square error (nRMSE) across a **wide range of 1D/2D/3D PDE problems**. Through extensive experiments, we show that explicit incorporation of fundamental physics knowledge significantly strengthens the generalization ability of neural operators. We will release models and codes at https://sites.google.com/view/sciml-fundemental-pde.

## 1 Introduction

Recent advances in scientific machine learning (SciML) have broadened traditional machine learning (ML) for modeling physical systems, using deep neural networks (DNNs) especially neural operators (NOs) (Li et al., 2021a; Pathak et al., 2022; Lam et al., 2023; Bi et al., 2023) as fast, accurate surrogates for solving partial differential equations (PDEs) (Raissi et al., 2019; Edwards, 2022; Kochkov et al., 2021; Pathak et al., 2022). However, compared to numerical methods, **a key disadvantage** of recent data-driven SciML models is their **limited integration of fundamental physical knowledge**.

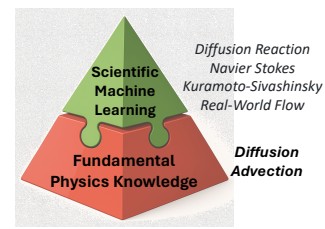

Figure 1: Can SciML models (e.g., neural operators trained on advanced PDEs) also understand fundamental physics knowledge (basic terms like diffusion, advection)?

Numerical solvers, though tailored to specific PDEs or discretizations, inherently preserve physical laws (e.g., conservation, symmetry), ensuring consistent and plausible simulations across diverse conditions (physical parameters, boundaries, geometries, etc.) (Ketcheson et al., 2012; Hansen et al., 2023; Mouli et al., 2024; Holl & Thuerey, 2024). In contrast, data-driven models, despite learning across PDE types (e.g., via multiphysics pretraining in SciML foundation models (McCabe et al., 2023; Hao et al., 2024)), remain sensitive to training distributions, degrading under distribution shifts (Subramanian et al., 2023; Benitez et al., 2024) and requiring large, diverse datasets. This fragility is worsened by the absence of rigorous verification: **Unlike classical solvers, SciML models are rarely evaluated against decomposed PDE components**. This gap introduces three major challenges: 1) **High data demands**: Without physics priors, neural operators require large, diverse datasets for high precision, as seen in recent SciML foundation models(Hao et al., 2024; McCabe

---

*Co-corresponding authors.

et al., 2023) which focus on generalization without addressing training data efficiency. 2) **Physical inconsistency**: Lacking inductive biases, these models may violate conservation laws or produce unphysical outputs, particularly in long-term rollout predictions. 3) **Poor generalization**: Neural PDE solvers often struggle with unseen simulation settings and requires retraining.

Motivated by the above challenges, we ask two scientific questions:

> **Q1**: Can neural operators **understand both** original PDEs and fundamental physics knowledge?
> **Q2**: Can neural operators benefit from **explicit learning** of fundamental physics knowledge?

In this paper, we highlight the importance of enforcing the learning of fundamental physical knowledge in neural operators. The key idea is to *identify physically plausible basic terms* that can be *decomposed from original PDEs*, and *incorporate their simulations during training*. Although often overlooked in SciML, our experiments demonstrate that these fundamental physical terms encode rich physical knowledge. Not only can they be utilized without incurring additional computational costs, but they also *widely offer substantial and multifaceted benefits*. This opens up a new door to improve the comprehensive generalization of neural operators with data efficiency.

We summarize our contributions below:

1. Through comprehensive evaluations of public SciML models, we observe a strong correlation between performance on original PDEs and basic PDE terms, highlighting the importance of fundamental physical knowledge in neural operators (Section 2.2).

2. We propose to explicitly incorporate fundamental physical knowledge into neural operators. Specifically, we design a simple and principled multiphysics strategy to train neural operators on simulations of both the original PDE and its basic form. (Section 3).

3. Our method exhibits three major benefits: 1) **data efficiency** (Section 4.2), 2) **long-term physical consistency** (Section 4.3), 3) **strong generalization in OOD (Section 4.4) and real-world (Section 4.5) scenarios**. We evaluate our method on a wide range of 1D/2D/3D PDEs, achieving consistent improvement in nRMSE (normalized root mean squared error, Section 4.2).

## 2 BACKGROUND

### 2.1 DEFINITION OF PDE LEARNING IN SCIML

For time-dependent PDEs, the solution is a vector-valued mapping $\boldsymbol{v} : \mathcal{T} \times \mathcal{S} \times \Theta \to \mathbb{R}^d$, defined on the temporal domain $\mathcal{T}$, the spatial domain $\mathcal{S}$, and the parameter space $\Theta$, with $d$ the number of dependent variables. Numerical solvers compute $\boldsymbol{v}_\theta(t, \cdot)$ from $\ell \geq 1$ past steps, enabling finite-difference approximations: $\mathcal{N}_\theta = [\boldsymbol{v}_\theta(t - \ell \cdot \Delta t, \cdot), \dots, \boldsymbol{v}_\theta(t - \Delta t, \cdot)] \mapsto \boldsymbol{v}_\theta(t, \cdot)$, where $\Delta t$ is the temporal resolution. SciML aims to learn a surrogate operator $\widehat{\mathcal{N}}_{\theta,\phi}$, parameterized by physical parameters $\theta$ and learnable weights $\phi$, that approximates this mapping. Given $N$ simulations, $\mathcal{D} := \{\boldsymbol{v}^{(i)}([0 : t_{\max}], \cdot) \mid i = 1, \dots, N\}$, the model is trained by minimizing a loss $L$, often the normalized root mean squared error: $\text{nRMSE} = \frac{\|\boldsymbol{v}_{\text{pred}} - \boldsymbol{v}\|_2}{\|\boldsymbol{v}\|_2}$, where $\boldsymbol{v}_{\text{pred}}$ is the model prediction.

### 2.2 MOTIVATION: NEURAL OPERATORS EXHIBIT CORRELATED YET WORSE PERFORMANCE ON FUNDAMENTAL PHYSICS

We begin with a motivating example to highlight the importance of incorporating fundamental physical knowledge into neural operators. Specifically, we gather publicly released pretrained neural operators, with a focus on *SciML foundation models* that are jointly pretrained across multiple PDEs ("multiphysics") (McCabe et al., 2023; Hao et al., 2024). We evaluate these models on their ability to capture fundamental physical knowledge (formally defined in Section 3.1), which none of them

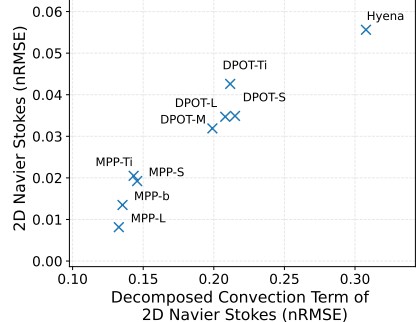

Figure 2: On 2D incompressible Navier-Stokes, neural operators and SciML foundation models (MPP (McCabe et al., 2023), DPOT (Hao et al., 2024), Hyena (Patil et al., 2023)) exhibit **correlated yet worse** performance on fundamental physics (x-axis).

were explicitly trained on, and compare their performance against the original PDE simulations.

From Figure 2, we observe a strong Pearson correlation (0.9625) between errors on original PDEs and their basic terms. This suggests that stronger SciML models implicitly learn basic PDE components more effectively. However, because these terms are not explicitly included in their training data, their absolute errors (0.133–0.308 on the x-axis) remain much larger than those for the original PDEs (0.008–0.056 on the y-axis). This gap reveals a key limitation: while the models demonstrate transfer across multiple physics, they still **fail to reliably capture the fundamental PDE components that underpin complex equations**. This motivates us to **explicitly** enforce an understanding of fundamental physical knowledge within neural operators.

## 3 METHODS

Motivated by our observation in Section 2.2, we incorporate fundamental physics knowledge into learning neural operators via: 1) Defining and decomposing basic forms from the original PDE (Section 3.1); 2) Jointly training neural operators on simulations from both basic forms and the original PDE (Section 3.2). We provide an overview of our method in Figure 3.

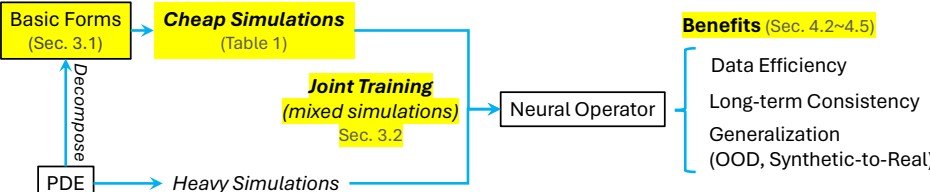

Figure 3: Overview of our method. Decomposed PDEs encode rich fundamental physical knowledge and introduce cheaper simulations. By jointly training on both the full PDE and its decomposed basic form, we bring multiple benefits to neural operators.

### 3.1 FUNDAMENTAL PHYSICAL KNOWLEDGE VIA DECOMPOSED BASIC PDE FORMS

#### 3.1.1 DEFINING FUNDAMENTAL PHYSICAL KNOWLEDGE OF PDES

Let us consider the second-order PDE as a general example:

$$\sum_{i,j=1}^{n} a_{ij}(\boldsymbol{x}, \boldsymbol{u}, \nabla\boldsymbol{u})\frac{\partial^2 \boldsymbol{u}}{\partial x_i \partial x_j} + \sum_{i=1}^{n} b_i(\boldsymbol{x}, \boldsymbol{u}, \nabla\boldsymbol{u})\frac{\partial \boldsymbol{u}}{\partial x_i} + c(\boldsymbol{x}, \boldsymbol{u}, \nabla\boldsymbol{u}) = f(\boldsymbol{x}), \tag{1}$$

where $\boldsymbol{u}$ is the target solution, with $\boldsymbol{x} \in \mathbb{R}^n$ the physical space (e.g., $n = 3$ for 2D time-dependent PDEs). The coefficients $a_{ij}, b_i, c$ ("physical parameters") govern the dynamics; mismatches between training and testing values cause domain shifts, leading to out-of-distribution (OOD) simulations. Finally, $f$ denotes an external forcing function (Nandakumaran & Datti, 2020).

We establish a systematic process to define the **fundamental physical knowledge of PDEs**, namely **basic PDE terms**: 1) retain terms that govern the essential and dominant physical dynamics; 2) remove terms that induce solver stiffness, increase computational cost, or contribute little to the pattern formation of interest. This procedure typically yields a simplified PDE form that can be simulated far more efficiently, while still capturing the key physical dynamics of the original system. From a machine learning perspective, decomposing a PDE into its basic form acts as a data augmentation strategy that reduces data collection costs.

Importantly, our approach of discovering and incorporating basic PDE forms differs fundamentally from the multiphysics training in recent SciML foundation models (McCabe et al., 2023; Hao et al., 2024), which simply aggregate diverse or even weakly related PDE systems. In contrast, we stress that neural operators must support basic PDE terms as a foundation while learning complex PDEs. Figure 4 illustrates the PDEs studied alongside their corresponding decomposed basic forms. An ablation study on different choices of terms is also provided in Appendix D.6.

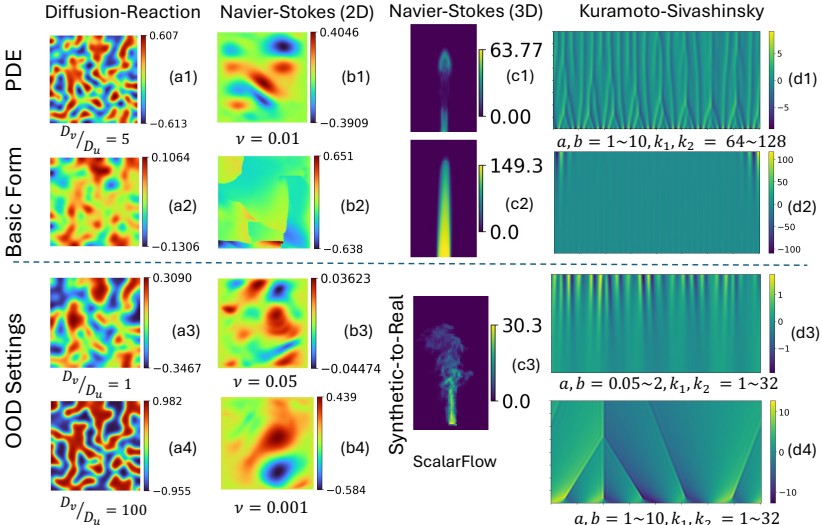

Figure 4: Visualizations of simulations of PDEs and their decomposed basic forms (Section 3.1). From left to right: Diffusion-Reaction (activator concentration), 2D Navier-Stokes (fluid velocity), 3D Navier-Stokes (smoke density), and Kuramoto-Sivashinsky (perturbation amplitude). Basic PDE forms are used for training neural operators with fundamental physics knowledge (Section 3.2), and the OOD settings are used for evaluating the generalization of neural operators (Section 4.4). $D_v, D_u$: diffusion coefficients (Equation 2). $\nu$: viscosity (Equation 4). $a, b, k_1, k_2$: magnitudes and wavenumbers (Equation 7).

### 3.1.2 DIFFUSION-REACTION

The Diffusion-Reaction equation models an activator-inhibitor system, which typically happens in the dynamics of chemistry, biology, and ecology. The Diffusion-Reaction equation describes spatiotemporal dynamics where chemical species or biological agents diffuse through a medium and simultaneously undergo local reactions. These systems are often used to model pattern formation, such as Turing patterns, in domains ranging from chemistry to developmental biology.

$$\partial_t u = D_u \partial_{xx} u + D_u \partial_{yy} u + R_u, \quad \partial_t v = D_v \partial_{xx} v + D_v \partial_{yy} v + R_v. \tag{2}$$

In Equation 2, $u$ and $v$ represent the concentrations of activator and inhibitor, respectively, and $D_u, D_v$ are diffusion coefficients. The nonlinear reaction terms $R_u, R_v$ model biochemical interactions. In our experiments, we adopt the FitzHugh–Nagumo variant with $R_u(u, v) = u - u^3 - k - v$ and $R_v(u, v) = u - v$, where $k = 5 \times 10^{-3}$, consistent with values used in models of excitable media such as neurons or cardiac tissue.

**Decomposed Basic Form.** To isolate the fundamental transport behavior and reduce simulation cost, we consider a simplified form of Equation 2 by omitting the nonlinear reaction terms $R_u$ and $R_v$, yielding pure diffusion equations:

$$\partial_t u = D_u \partial_{xx} u + D_u \partial_{yy} u, \quad \partial_t v = D_v \partial_{xx} v + D_v \partial_{yy} v. \tag{3}$$

- *Why Drop Reaction Terms?* This form retains the essential dispersal dynamics but eliminates the feedback coupling between $u$ and $v$. Nonlinear reaction terms can vary rapidly, introducing stiffness into the PDE. This stiffness necessitates smaller time steps for stable numerical integration, increasing computational cost. By omitting these nonlinear terms, the system becomes linear and more amenable to efficient numerical solutions.

- *Why Prioritize the Diffusion Term?* Pure diffusion, though simpler, encodes key properties such as isotropic spreading and mass conservation, providing inductive bias for learning. Unlike reaction terms, which act locally to update activator and inhibitor concentrations, the diffusion term governs spatial coupling, as the primary source of pattern formation and spatial dynamics, facilitating transport and stabilization, which explains the visually similar patterns in Figures 4 a1 and a2.

**Physical Scenarios.** The emergence of spatial patterns in reaction-diffusion systems is governed by the ratio $\frac{D_v}{D_u}$, which affects the relative spreading rates of inhibitor and activator (Page et al.,

2005; Asgari et al., 2011; Gambino et al., 2024). Classical Turing instability arises when $D_v \gg D_u$, leading to diffusion-driven pattern formation, and the inhibitor spreads out while the activator stays localized. Following previous works (Menou et al., 2023), we set $D_u = 1 \times 10^{-3}$, and focus on learning simulations when $\frac{D_v}{D_u} = 5$, and possible OOD scenarios when $\frac{D_v}{D_u} \in \{1, 100\}$.

### 3.1.3 INCOMPRESSIBLE NAVIER-STOKES

The Navier–Stokes equations govern the dynamics of fluid flow and serve as fundamentals for fluid simulations. It considers both the mass conservation and the momentum of fluid parcels.

$$\frac{\partial \boldsymbol{u}}{\partial t} = -(\boldsymbol{u} \cdot \nabla)\boldsymbol{u} + \nu \nabla^2 \boldsymbol{u} - \frac{1}{\rho}\nabla p + \boldsymbol{f}, \tag{4}$$

where $\boldsymbol{u}$ is the velocity field, $\nu$ is the dynamic viscosity, $\rho$ is the fluid density, $p$ is the fluid pressure, and $\boldsymbol{f}$ is the external force field.

**Decomposed Basic Form.** To isolate fundamental nonlinear transport mechanisms and reduce computational complexity, we simplify Equation 4 by omitting the pressure term (incompressibility via projection) $\frac{1}{\rho}\nabla p$ and diffusion term $\nu \nabla^2 \boldsymbol{u}$:

$$\frac{\partial \boldsymbol{u}}{\partial t} = -(\boldsymbol{u} \cdot \nabla)\boldsymbol{u} + \boldsymbol{f}, \tag{5}$$

This form captures inertial advection with external forcing and approximates high Reynolds number flows, where viscous effects are negligible. Such simplifications are analytically meaningful and are often used in turbulence modeling (e.g., inviscid Euler equations). Learning this reduced dynamics can help models internalize convection-dominant regimes.

- *Why Drop Pressure and Diffusion Terms?* The pressure term, which enforces fluid incompressibility, requires solving large linear systems and is difficult to parallelize and computationally expensive. Omitting it significantly accelerates the simulation. Similarly, the diffusion term in Navier-Stokes often uses explicit Euler integration with substeps, adding complexity. Removing it simplifies the simulation further. Moreover, for many visual effects like smoke or fire, viscosity is minimal, so the diffusion term has little visual impact and can often be omitted without noticeable loss in realism.

- *Why Prioritize the Convection Term?* Computationally, the convection term is cheap as it describes the local transport of fluid and there is no need to iterate across the spatial domain. Meanwhile, convection is the main driver of motion in most fluid flows, as it transports vorticity and mass. Without it, the fluid would just sit still or respond passively to forces. It captures nonlinear self-interaction, which is critical for dynamic, complex-looking behavior.

**Physical Scenarios.** In Navier-Stokes, the dynamic viscosity $\nu$ in Equation 4 (or the Reynolds number $Re = \frac{\rho u L}{\nu}$ where $\rho$ is the density of the fluid, $u$ is the flow speed, $L$ is the characteristic linear dimension) controls the fluid dynamics. It measures the balance between inertial forces (pushes the fluid particles in different directions, leading to chaotic flow patterns) and viscous forces (resists motion and smoothes out differences in velocity, promoting an orderly flow) of a fluid. Following previous works (Schlichting & Gersten, 2016; Kochkov et al., 2021; Page et al., 2024), we will mainly focus on learning simulations when $\nu = 0.01$, and possible OOD scenarios when $\nu \in \{0.05, 0.001\}$. Note that a smaller $\nu$ will lead to more turbulent flows.

**3D Extension.** In real-world scenarios such as atmospheric or smoke dynamics, buoyancy-driven flows provide additional complexity (Eckert et al., 2019). We extend our setting to simulate 3D incompressible Navier–Stokes in a rising plume scenario (see Figure 4 c1 and c3). We simulate how a plume of smoke rises and spreads in a 3D box-shaped environment. Smoke is introduced from a small circular inflow region located at the bottom center of the domain, at a steady inflow rate of 0.2 units per timestep. The smoke is carried upward due to buoyancy. This setting tests the method's robustness on complex spatiotemporal dynamics in three dimensions.

### 3.1.4 KURAMOTO-SIVASHINSKY

The Kuramoto-Sivashinsky equation is a nonlinear PDE that models the interplay of instability, nonlinearity, and dissipation, making it a prototype for studying spatiotemporal chaos. It is used to

simulate phenomena such as wrinkled flame fronts, thin fluid film instabilities, and chaotic pattern formation. We consider the following one-dimensional Kuramoto-Sivashinsky equation:

$$\frac{\partial u}{\partial t} = -u\partial_x u - \partial_{xx} u - \partial_{xxxx} u, \quad \text{on } [0, L] \times [0, T] \tag{6}$$

The solution $u$ generally represents the perturbation amplitude of a chaotic system. The spatial domain $[0, L]$ is equipped with periodic boundary conditions. We impose the initial condition on $x \in [0, L]$ as

$$u_0(x) = a\cos\left(\frac{k_1 \pi x}{L}\right) + b\cos\left(\frac{k_2 \pi x}{L}\right) + \sigma\epsilon(x), \tag{7}$$

where $u_0$ is the superposition of two cosine modes and small mean-zero Gaussian perturbations. This initialization, common in KS studies (Papageorgiou & Smyrlis, 1991; Gudorf & Cvitanovic, 2019), combines deterministic cosine modes that inject controlled perturbations with small random noise. $a, b, k_1, k_2$ are tunable physical parameters controlling the strength and decay of the initial perturbation. Here, $\epsilon(x)$ are i.i.d. standard normal samples, and we set $\sigma = 0.05$. Spatial derivatives are evaluated spectrally via discrete Fourier transforms, and time integration is performed with a fourth–order Runge–Kutta method.

**Decomposed Basic Form.** To isolate the linear stabilizing and destabilizing mechanisms and reduce computational complexity, we simplify the Kuramoto–Sivashinsky equation by omitting the nonlinear advection term $-u\partial_x u$. The reduced equation is

$$\frac{\partial u}{\partial t} = -\partial_{xx} u - \partial_{xxxx} u, \quad \text{on } [0, L] \times [0, T], \tag{8}$$

which captures the competition between the destabilizing anti-diffusion term ($-\partial_{xx} u$) and the stabilizing diffusion term ($-\partial_{xxxx} u$).

- *Why Drop the Nonlinear Advection Term?* For small amplitudes or short times, $-\boldsymbol{u}\partial_x \boldsymbol{u}$ is higher order in $\boldsymbol{u}$, so the linearized dynamics govern the instabilities and pattern formation. Removing this term also eliminates costly Fourier transforms, yielding a significant speedup in simulation.

- *Why Prioritize High-Order Stabilizing/Destabilizing Terms?* The balance between destabilizing $-\partial_{xx} u$ and stabilizing $-\partial_{xxxx} u$ determines how fast the chaotic model grows or decays. The fourth-order dissipation is particularly important for controlling stiffness and ensuring smoothness of solutions, making it critical for stable numerical integration. Finally, in practice, many interface/turbulence models reduce to precisely this "anti-diffusion + diffusion" structure, so analyzing these terms provides broadly transferable insights.

**Physical Scenarios.** In the initial condition (Equation 7), the amplitudes $a, b$ control the strength of nonlinearity, while wavenumbers $k_1, k_2$ control the growth or decay of chaos, yielding regimes 1) For large $a, b$ and high $k_1, k_2$, the dynamics are predominantly **linear**, and chaos decays over time. 2) For small $a, b$ and small $k_1, k_2$, the system develops **weakly nonlinear** chaos, with instabilities growing slowly. 3) For large $a, b$ and small $k_1, k_2$, the system exhibits **strongly nonlinear** chaotic growth. In our experiments, we focus on learning in the linear regime with $a, b \in [1, 10]$ and $k_1, k_2 \in [64, 128]$, and evaluate on two out-of-distribution (OOD) regimes: weak nonlinearity ($a, b \in [0.05, 0.2]$, $k_1, k_2 \in [1, 32]$) and strong nonlinearity ($a, b \in [1, 10]$, $k_1, k_2 \in [1, 32]$).

## 3.2 Joint Learning with Fundamental Physical Knowledge

After defining our fundamental physics knowledge, we now explain how to integrate it into learning neural operators in principle from two perspectives: data composition and neural architecture.

**1) Data Composition.** We jointly train neural operators on simulations of both our PDE and the decomposed basic form as a multiphysics training with a composite dataset. We summarize our simulations in Table 1. Since the simulation costs of decomposed basic forms are much cheaper than the original PDE, we can "trade-in" simulations of the original PDE for more simulations of basic forms **under the same simulation costs**. We define the "Sample Mixture Ratio" as the rate ***derived*** from the rate of simulation costs of original PDE with its decomposed basic form while making sure

Table 1: Summary of simulations of PDEs and their decomposed basic forms. GPU: NVIDIA RTX 6000 Ada.

| PDE | Spatial Resolution | Temporal Steps | Target Variables | Simulation Costs (sec. per step) | Sample Mixture Ratio (PDE : Basic Form) |
|---|---|---|---|---|---|
| Diffusion-Reaction (Eq. 2) Basic Form (Eq. 3) | $128 \times 128$ | 100 | Activator $u$, Inhibitor $v$ | $1.864 \times 10^{-2}$ $6.610 \times 10^{-3}$ | 1:3 |
| Navier-Stokes (2D) (Eq. 4) Basic Form (2D) (Eq. 5) | $256 \times 256$ | 1000 | Velocity $\boldsymbol{u}$, Density $s$ | 2.775 0.113 | 1:24 |
| Navier-Stokes (3D) (Eq. 4) Basic Form (3D) (Eq. 5) | $50 \times 50 \times 89$ | 150 | Velocity $\boldsymbol{u}$, Density $s$ | 1.047 0.300 | 1:3 |
| Kuramoto-Sivashinsky (Eq. 6) Basic Form (Eq. 8) | 512 | 200 | Perturbation Amplitude $u$ | $1.176 \times 10^{-4}$ $9.135 \times 10^{-6}$ | 1:12 |

that "exchange" one primary data with the corresponding number of basic form data will **maintain a comparable or reduced simulation budget when training baseline and our proposed model**.

We apply a multi-task formulation, where the model learns from both the original PDE and its simplified basic form. The idea is inspired from curriculum learning (Bengio et al., 2009; Pentina et al., 2014) and auxiliary task learning (Liu et al., 2019). In our method, the basic forms act as a simpler, physically motivated auxiliary task that can facilitate more efficient representation learning and accelerate the convergence on the primary task.

**2) Neural Architecture.** We mainly consider the Fourier Neural Operator (FNO) (Li et al., 2021a) as our neural architecture. However, we make our method agnostic to specific architectures of neural operators: We generally share the backbone of the neural operator for learning both the main PDE and its basic terms, while employing separate final prediction layers for the two tasks. We will discuss the details on model structure in Appendix B, and will include more results using transformer (Dosovitskiy et al., 2020; Tong et al., 2022; McCabe et al., 2023; Chen et al., 2024) in Appendix D.

## 4 EXPERIMENTS

### 4.1 SETTINGS

Our baseline is learning the original PDE problem. In general, our method reallocates half of the baseline's simulation budget to simulate the basic PDE terms, with the sample mixture ratio defined in Table 1. To fairly compare with the baseline, we adopt the same **hyperparameters and optimization costs (number of gradient descent steps)**. Since our goal is to evaluate performance on the original PDE, we use data from the basic PDE term only during training. All testing is conducted exclusively on the original PDE data with 100 samples (Takamoto et al., 2022). We use Adam optimizer, cosine annealing learning rate scheduler, and nRMSE defined in Section 2.1. We summarize our training details in Appendix C.

### 4.2 DATA EFFICIENCY

We first study our method with different numbers of training samples, and demonstrate that neural operators trained with our method can achieve stronger performance with less training data. We consider the following methods:

- "Baseline": Neural operators that are only trained on simulations of the original PDE.

- Ours: As described in Section 3.2, we can replace simulations of the original PDE with its decomposed basic form, allowing the total sim-

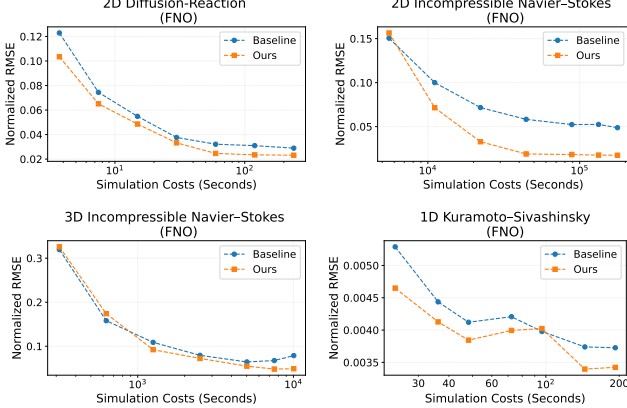

Figure 5: Joint training neural operators on data of the original PDE and the basic form improves performance and data efficiency. Y-axis: normalized RMSE. X-axis: simulation costs (seconds).

ulation cost of the training data to be comparable or even reduced. See Table 1 for the sample mixture rate for fair comparison.

In Figure 5, we study prediction errors of neural operators trained with different numbers of simulations (measured in their simulation costs). We can see that, across PDEs and neural architectures, our method (orange square) is to the lower left of the baseline (blue circle), which means that we can achieve *improved prediction errors with reduced simulation costs*.

## 4.3 LONG-TERM PHYSICAL CONSISTENCY

Next-frame prediction (Takamoto et al., 2022; McCabe et al., 2023; Hao et al., 2024) is a widely adopted evaluation, where input frames are always ground truth. Meanwhile, autoregressive inference, where the model keeps rolling out to further temporal steps with its own output as inputs, is a meaningful and more challenging stress test. In autoregressive inference, a model forecasts futures with its own (noisy) output as inputs, and thus prediction error will accumulate along rollout steps. We can see that our improvements in Figure 5 further persist across five autoregressive steps in Figure 6, leading to improved long-term consistency.

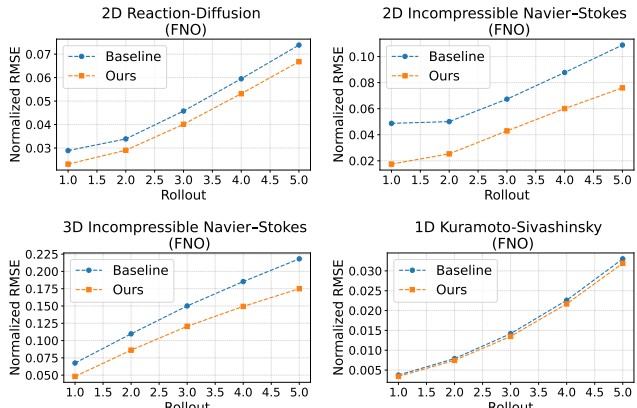

Figure 6: Joint training neural operators on data of the original PDE and the basic form improves performance with autoregressive inference at different unrolled steps. Models are evaluated using the best-performing checkpoints from training, shown in Figure 5.

## 4.4 OUT-OF-DISTRIBUTION (OOD) GENERALIZATION

We next show the benefits of our method towards the generalization of neural operators in out-of-distribution (OOD) settings, where the physical parameters used during simulation are significantly shifted. We consider physical scenarios described in Section 3.1, and show results in Table 2. We can see that our method not only improves in-distribution errors, but also generalizes better on unseen physical dynamics (simulations by unseen parameters), leading to more robust neural operators.

Table 2: Comparisons of OOD generalization for different training methods. Models are evaluated using the best-performing checkpoints from training, as shown in Figure 5, under comparable simulation cost settings.

| PDE | Model | Source | | Target 1 | | Target 2 | |
|---|---|---|---|---|---|---|---|
| | | Setting | nRMSE | Setting | nRMSE | Setting | nRMSE |
| Diffusion-Reaction (2D) | Baseline | $\frac{D_v}{D_u}=5$ | 0.0289 | $\frac{D_v}{D_u}=1$ | 0.0413 | $\frac{D_v}{D_u}=100$ | 0.0770 |
| | Ours | | **0.0231** | | **0.0331** | | **0.0538** |
| Navier-Stokes (2D) | Baseline | $\nu=0.01$ | 0.0487 | $\nu=0.05$ | 0.0825 | $\nu=0.0001$ | 0.0369 |
| | Ours | | **0.0175** | | **0.0222** | | **0.0125** |
| Navier-Stokes (3D) | Baseline | $\nu=0.01$ | 0.0675 | $\nu=0.1$ | 0.0393 | $\nu=0.0001$ | 0.0836 |
| | Ours | | **0.0481** | | **0.0329** | | **0.0602** |
| Kuramoto-Sivashinsky (1D) | Baseline | a, b = $1\sim10$ $k_1, k_2 = 64\sim128$ | 0.0037 | a, b = $0.05\sim2$ $k_1, k_2 = 1\sim32$ | 0.0021 | a, b = $1\sim10$ $k_1, k_2 = 1\sim32$ | 0.0200 |
| | Ours | | **0.0034** | | **0.0018** | | **0.0197** |

## 4.5 SYNTHETIC-TO-REAL GENERALIZATION

Finally, we test neural operators trained on simulations of 3D Navier-Stokes in real-world scenarios. Essentially, transferring models trained on simulations to real observations is a synthetic-to-real generalization problem (Chen et al., 2020; 2021), as domain gaps between numerical simulations and real-world measurements always persist. We study the ScalarFlow dataset (Eckert et al., 2019), which is a reconstruction of real-world smoke plumes and assembles the first large-scale dataset of realistic turbulent flows. We provide visualizations of synthetic and ScalarFlow data in Figure 4.

We show the results and visualize the ground truth as well as the model predictions on smoke plumes from ScalarFlow in Figure 7. We can see that our method outperforms the baseline model and presents a qualitative comparison of scalar flow predictions on real data, illustrating that our jointly trained model exhibits improved synthetic-to-real generalization performance. Please read our Appendix D.2 for more results on 3D Navier-Stokes.

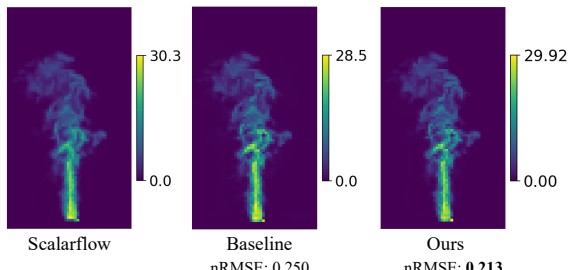

Figure 7: Visualizations of the last time step in the ScalarFlow and its predictions derived by baseline and our model.

## 5 RELATED WORKS

**Machine Learning for Scientific Modeling** Learning-based methods have long been used to model physical phenomena (Lagaris et al., 1998; 2000; Chen & Chen, 1995b;a). Physics-informed neural networks (PINNs) (Raissi et al., 2019; Zhu et al., 2019; Geneva & Zabaras, 2020; Gao et al., 2021; Ren et al., 2022) incorporate PDEs into loss functions to enforce physical laws, but often struggle with generalization and optimization issues (Krishnapriyan et al., 2021; Edwards, 2022). Operator learning methods like Fourier Neural Operators (FNO) (Li et al., 2021a; 2020; Kovachki et al., 2023) and DeepONet (Lu et al., 2019) offer greater flexibility by learning mappings between function spaces but require extensive labeled data (Raissi et al., 2019; Brandstetter et al., 2022; Zhang et al., 2023; Chen et al., 2024; Liu et al., 2026). We adopt a unique approach by evaluating and enhancing SciML through the lens of its compatibility with fundamental physical principles.

**Data-Driven Neural PDE Solvers** Machine learning has increasingly been used to approximate PDE solutions, with neural PDE solvers trained on diverse scenarios to mimic traditional simulators. Early models used CNNs (Guo et al., 2016; Zhu & Zabaras, 2018; Bhatnagar et al., 2019), while DeepONet (Lu et al., 2019) introduced a neural operator (NO) framework separating input and query encodings, inspiring many extensions (Wang et al., 2021; Hadorn, 2022; Wang et al., 2022; Lin et al., 2023; Venturi & Casey, 2023; Xu et al., 2023; McCabe et al., 2023; Hao et al., 2024). Advances like FNO (Li et al., 2021a), LNO (Cao et al., 2023), CNO (Raonic et al., 2024), and KNO (Xiong et al., 2024) have expanded the field, with FNO particularly impactful across applications (Li et al., 2021b; Guibas et al., 2021; Yang et al., 2021; Rahman et al., 2022b;a; Pathak et al., 2022; Liu et al., 2022). Compared with previous neural operator works, instead of naively swapping in cheaper simulations of simplified PDEs, the core merit of our work is to emphasize the multifaceted benefits of explicit learning of fundamental physics knowledge during operator learning.

**Out-of-Distribution Generalization in SciML** Interest in out-of-distribution (OOD) generalization for scientific machine learning (SciML) has grown recently. Subramanian et al. (2023) showed that fine-tuning neural operators (NOs) on OOD PDEs often requires many OOD simulations, which may be impractical. Benitez et al. (2024) proposed a Helmholtz-specific FNO with strong OOD performance, supported by Rademacher complexity and a novel risk bound. Other work includes ensemble methods leveraging uncertainty (Hansen et al., 2023; Mouli et al., 2024), loss functions informed by numerical schemes (Kim & Kang, 2024), and meta-learning for varied geometries (Liu et al., 2023). However, varying PDE types and setups across studies hinder unified insights into OOD generalization for NOs. Our work demonstrates that neural operators explicitly trained with fundamental physics knowledge exhibit improved OOD and synthetic-to-real generalization.

## 6 CONCLUSION

We present a principle and architecture-agnostic approach to enhance neural operators by explicitly incorporating fundamental physical knowledge into their training. By decomposing complex PDEs into simpler, physically meaningful basic forms and using them as auxiliary training signals, our proposed method significantly improves data efficiency, long-term predictive consistency, and out-of-distribution generalization. These improvements are demonstrated across a variety of PDE systems and neural operator architectures. Our finding highlights the untapped potential of fundamental

physics as an inductive bias in scientific machine learning, offering a robust and cost-effective pathway to more reliable and generalizable surrogate models in real-world physical simulations.

## THE USE OF LARGE LANGUAGE MODELS (LLMS)

LLMs did *not* play a significant role in either the research ideation or the writing of this paper. Their use was limited to correcting minor grammatical issues and typographical errors.

## ACKNOWLEDGMENT

This research used resources of the National Energy Research Scientific Computing Center, a DOE Office of Science User Facility supported by the Office of Science of the U.S. Department of Energy under Contract No. DE-AC02-05CH11231 using NERSC award NERSC DDR-ERCAP0034682 and ASCR-ERCAP0031463. J. Cao's research is partially supported by the Discovery Grants (RGPIN-2023-04057) of the Natural Sciences and Engineering Research Council of Canada (NSERC) and the Canada Research Chair program.

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

# A  SIMULATION SETTINGS

In this section, we detail the simulation settings for the 2D Diffusion-Reaction, 2D and 3D Incompressible Navier-Stokes and 1D Kuramoto–Sivashinsky (see Table 1 for the summary). We will also explain how we prepare simulations for the "Spatiotemporal Downsampling" in Section D.1.

To ensure a fair comparison under equivalent simulation costs with the basic form of each PDE, we downsample the original PDE simulations both spatially and temporally. We also introduce the settings for the corresponding reduced spatiotemporal resolution simulations. Note that the simulation cost of these downsampled settings is matched to that of the basic form, which implies that their sample mixture ratios in joint training remain equivalent.

To further explore the benefits of our multiphysics joint training approach with this reduced spatiotemporal resolution simulation strategies (refer to *Ours@Spatiotemporal* in Figure 9), we additionally introduce the simulation settings for the spatiotemporally downsampled basic form of the 2D Diffusion-Reaction equation. As we will discuss in Section D.1, our framework is orthogonal to standard downsampling techniques, and combining the two can lead to further reductions in simulation cost. This reduction allows for an increased proportion of basic form samples in the training mixture under a fixed computational budget (see Table 3 for details).

## A.1  DIFFUSION-REACTION

Our simulation setting for Diffusion-Reaction follows (Takamoto et al., 2022). Our solver is Py-Claw (Ketcheson et al., 2012) that uses the finite volume method. We set the initial condition as standard normal random noise $u(t = 0, x, y) \sim \mathcal{N}(0, 1.0)$. We use the homogeneous Neumann boundary condition. We simulate in a spatial domain of $\Omega = [-1, 1]^2$, with resolution $128 \times 128$. We simulate 5 seconds and save into 100 temporal steps.

**Reduced Spatiotemporal Resolution.**  When simulating the original Diffusion-Reaction equation at low spatiotemporal grids (yellow curves in Figure 9), we reduce the spatial resolution from $128 \times 128$ to $96 \times 96$, and reduce the number of temporal steps from 100 to 50. We then upsample to $128 \times 128 \times 100$ (steps) via bilinear interpolation to match the resolution of simulations of the original PDE. The total simulation interval is maintained at 5 seconds, preserving the underlying physical dynamics.

Similarly, we can further simulate our decomposed basic form of Diffusion-Reaction at low spatiotemporal resolution (green curve in Figure 9). Table 3 shows the simulation cost of decomposed basic forms with reduced spatiotemporal resolution and the sample mixture ratio.

Table 3: Summary of 2D Diffusion-Reaction simulation and its decomposed basic forms with reduced spatiotemporal resolution. "Sample Mixture Rate": We replace simulations of the original PDE with its decomposed basic form with reduced spatiotemporal resolution and make sure the total simulation cost of the training data can be comparable. GPU: NVIDIA RTX 6000 Ada.

| PDE | Spatial Resolution | Temporal Steps | Target Variables | Simulation Costs (sec. per step) | Sample Mixture Ratio (PDE : Basic Form) |
|---|---|---|---|---|---|
| Diffusion-Reaction (Eq. 2) | $128 \times 128$ | 100 | | $1.864 \times 10^{-2}$ | |
| Basic Form (Eq. 3) with Reduced Spatiotemporal Resolution | $96 \times 96$ | 50 | Activator $u$, Inhibitor $v$ | $2.390 \times 10^{-3}$ | 1:8 |

## A.2  2D INCOMPRESSIBLE NAVIER-STOKES

Our simulation setting for incompressible Navier-Stokes follows (Takamoto et al., 2022). Our solver is PhiFlow (Holl & Thuerey, 2024). We simulate in a spatial domain of $\Omega = [0, 1]^2$, with resolution $256 \times 256$. We simulate 5 seconds with a $dt = 5 \times 10^{-5}$, and periodically save into 1000 temporal steps. Our initial conditions $\boldsymbol{u}_0$ and forcing term $\boldsymbol{f}$ are drawn from isotropic Gaussian random fields, where the low-frequency components of the spectral density are scaled with `scale` and high-frequency components are suppressed with power-law decay by `smoothness`. For $\boldsymbol{u}_0$, `scale` is 0.15 and `smoothness` is 3.0. For $\boldsymbol{f}$, `scale` is 0.4 and `smoothness` is 1.0. Boundary conditions are Dirichlet.

**Reduced Spatiotemporal Resolution.** When simulating the original 2D incompressible Navier-Stokes equation at low spatiotemporal grids (yellow curves in Figure 9), we reduce the spatial resolution from $256 \times 256$ to $100 \times 100$. We then spatially upsample to $256 \times 256$ via bilinear interpolation to match the resolution of simulations of the original PDE. To reduce the temporal resolution while maintaining the same total simulation time and number of recorded frames, we increase the time-step size and proportionally reduce the number of integration steps and output interval. Specifically, we change the time-step from $dt = 5 \times 10^{-5}$ to $dt = 5 \times 10^{-4}$, and reduce the total number of time steps from $n_{\text{steps}} = 100{,}000$ to $n_{\text{steps}} = 10{,}000$. To preserve the temporal spacing between output frames, we decrease the frame interval from 100 to 10. This ensures the same total simulation duration of 5 seconds and the same number of output frames (1,000). This modification reduces computational cost by roughly 10 times.

## A.3 3D INCOMPRESSIBLE NAVIER-STOKES

Our solver is PhiFlow (Holl & Thuerey, 2024). We simulate in a spatial domain of $\Omega = [0, 1]^3$, with resolution $50 \times 50 \times 89$. We simulate 150 steps with a $dt = 2 \times 10^{-4}$. We set the initial $\boldsymbol{u}_0$ as zero and upward buoyancy forcing term $\boldsymbol{f}_z = 5 \times 10^{-4}$. Unlike the 2D Navier-Stokes, we introduce randomness of the buoyancy forcing term on horizontal directions by uniformly drawing $\boldsymbol{f}_x \, \boldsymbol{f}_x$ from $[-1, 1] \times 10^{-4}$. We set Dirichlet zero boundary conditions.

## A.4 1D KURAMOTO–SIVASHINSKY

Our simulation setting for the one-dimensional Kuramoto–Sivashinsky equation follows (Papageorgiou & Smyrlis, 1991; Gudorf & Cvitanovic, 2019). Our solver is a pseudospectral exponential time-differencing fourth-order Runge–Kutta scheme, implemented with 64 contour points. We use periodic boundary conditions on the spatial domain $\Omega = [0, L]$ with $L = 64\pi$, discretized with equispaced $s = 512$ grid points. Spatial derivatives are evaluated spectrally via FFT (fast Fourier transform), and nonlinear terms are treated in physical space through pseudospectral squaring and differentiation. We simulate up to the final time $T = 30$, using an internal time step $dt = 0.1$, and record $n_{\text{steps}} = 200$ snapshots uniformly in time.

## B MODEL STRUCTURE

We mainly consider the Fourier Neural Operator and the transformer in this study. The model structures are shown in Figure 8. The basic form and the original PDE share all layers but the last prediction layer, which is agnostic to specific architectures of neural operators. This task-specific output layer is a very well-known and widely adopted configuration in multi-task learning. It has been highlighted in survey papers that hard parameter sharing (one input, shared hidden layers, multiple outputs) is the standard setup for multi-task learning due to its efficiency and representational benefits (Ruder, 2017; Crawshaw, 2020).

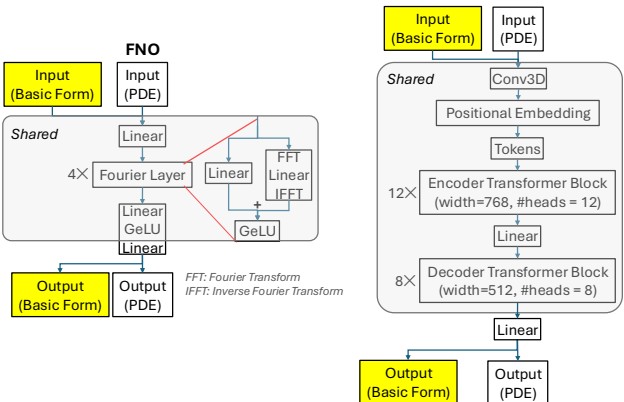

Figure 8: Our method is agnostic to specific architectures of neural operators: we always share the backbone of the model between learning the original PDE and its basic form, and decouple their predictions in the last layer.

## C MORE IMPLEMENTATION DETAILS

We summarize our training details in Table 4. We conducted our experiments on NVIDIA RTX 6000 Ada GPUs, each with 48 GB of memory.

Table 4: Training details. "DR": Diffusion-Reaction."NS": Navier-Stokes."KS": Kuramoto–Sivashinsky.

| | 2D DR (FNO) | 2D DR (Transformer) | 2D NS (FNO) | 2D NS (Transformer) | 3D NS (FNO) | 3D NS (Transformer) | 1D KS (FNO) |
|---|---|---|---|---|---|---|---|
| Input Shape Format | $H \times W \times T \times C\ (C = 2)$ | | $H \times W \times T \times C\ (C = 3)$ | | $X \times Y \times Z \times T \times C\ (C = 4)$ | | $T \times S$ |
| Number of Training Samples (PDE Simulations) | 2, 4, 8, 16, 32, 64, 128 | | 2, 4, 8, 16, 32, 48, 64 | | 2, 4, 8, 16, 32, 48, 64 | | $1024 \sim 8192$ |
| Input Time Steps ($\ell$ in Section 2.1) | 10 | 10 | 10 | 10 | 10 | 10 | 10 |
| Sample Mixture Ratio | 1:3 | 1:3 | 1:24 | 1:6 | 1:3 | 1:3 | 1:12 |
| Learning Rate | 0.001 | 0.0003 | 0.001 | 0.001 | 0.001 | 0.00015 | 0.001 |
| Batch Size for Primary Data | 4 | 8 | 16 | 16 | 8 | 8 | 64 |
| Epochs | 100 | 60 | 20 | 30 | 20 | 80 | 50 |
| Auxiliary Task Loss Weight | 0.7 | 0.7 | 0.7 | 0.7 | 0.7 | 0.7 | 0.7 |
| Training Hours | $0.08 \sim 1.83$ | $0.6hr \sim 7hr$ | $1 \sim 29$ | $1.5 \sim 45$ | $0.5 \sim 6.5$ | $16 \sim 120$ | $2 \sim 18$ |
| Gradient Descent Steps Per Epoch (Baseline and Ours) | $46 \sim 2912$ | $23 \sim 1456$ | $124 \sim 3960$ | $124 \sim 3960$ | $70 \sim 2240$ | $70 \sim 2240$ | $3040 \sim 24320$ |

# D   MORE RESULTS

Beyond the main comparison between the baseline and our proposed model using FNO showed in Section 4, we further conducted additional experiments to assess our approach. In this section, we will show the results from both FNO and Transformer.

## D.1   COMPARISON OF DATA EFFICIENCY AND OUT-OF-DISTRIBUTION GENERALIZATION AGAINST SPATIOTEMPORAL DOWNSAMPLING

To demonstrate the effectiveness of our method, we conducted an ablation study comparing it with Spatiotemporal Downsampling on both 2D Diffusion–Reaction (DR) and 2D Navier–Stokes (NS). We defined the Spatiotemporal Downsampling method as follows: Neural operators that are trained with a mixture of simulations of the original PDE and simulations at low spatial and temporal resolutions (then linearly interpolated to the original resolution). Similar to our method, we can also save simulation costs with reduced spatiotemporal resolutions. See Appendix A for rates of downsampling and more details. Meanwhile, as our decomposed basic form is orthogonal to spatiotemporal downsampling during simulations, our method can serve as a complementary data augmentation. On 2D Diffusion-Reaction, we can simulate our decomposed basic forms at lower spatiotemporal resolution, leading to further reduced simulation costs and improved data efficiency (green triangle), outperforming the baseline at the lower spatiotemporal resolution (yellow diamond). For 2D Navier–Stokes with the Transformer, our method still outperforms both the baseline and spatiotemporal downsampling at comparable cost.

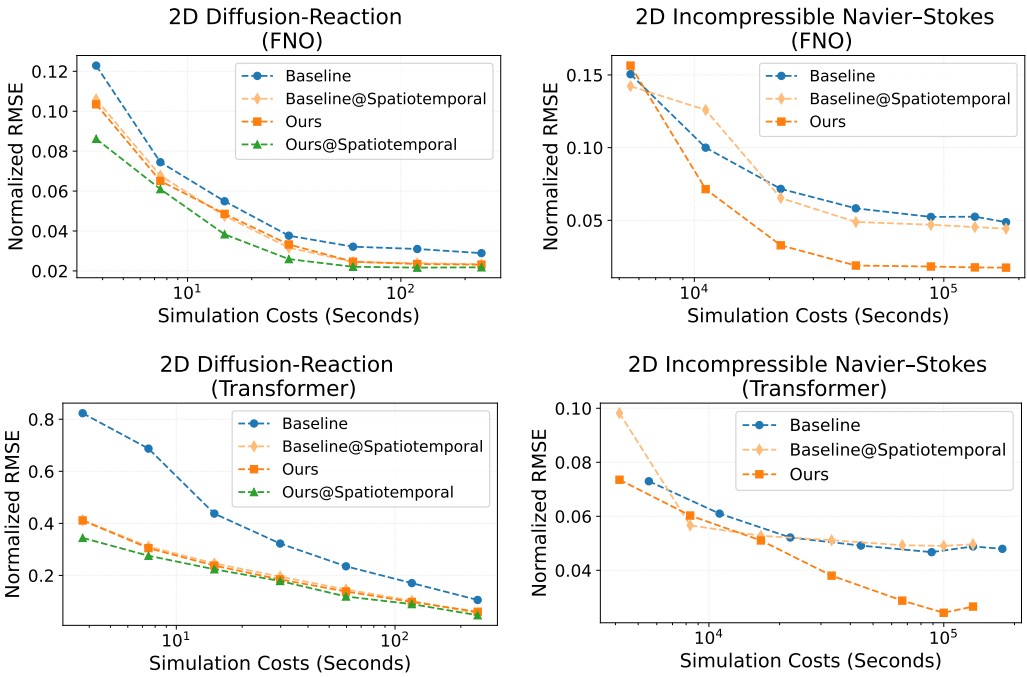

Figure 9: Joint training neural operators on data of the original PDE and the basic form improves performance and data efficiency. "Spatiotemporal": short for "Spatiotemporal Downsampling". Y-axis: normalized RMSE. X-axis: simulation costs (seconds).

Table 5 reports the out-of-distribution (OOD) generalization results across both the 2D Diffusion-Reaction and Navier-Stokes equations. Similar to the results in Table 2, here we can see that our approach not only improves in-distribution errors but also consistently enhances generalization to simulations of unseen physical parameters. This robustness holds across both FNO and Transformer architectures, leading to more reliable and consistent neural operators under varying conditions.

Table 5: Comparisons of OOD generalization for different training methods with the Transformer. Models are evaluated using the best checkpoints from training in Figure 5, under comparable simulation cost settings. "Spatiotemporal": short for "Spatiotemporal Downsampling".

| PDE | Model | Source Setting | Source nRMSE | Target 1 Setting | Target 1 nRMSE | Target 2 Setting | Target 2 nRMSE |
|---|---|---|---|---|---|---|---|
| Diffusion-Reaction (2D, FNO) | Baseline | | 0.0289 | | 0.0413 | | 0.0770 |
| | Baseline@Spatiotemporal | $\frac{D_v}{D_u}=5$ | 0.0234 | $\frac{D_v}{D_u}=1$ | 0.0303 | $\frac{D_v}{D_u}=100$ | 0.0663 |
| | Ours | | 0.0231 | | 0.0331 | | **0.0538** |
| | Ours@Spatiotemporal | | **0.0218** | | **0.0298** | | 0.0596 |
| Diffusion-Reaction (2D, Transformer) | Baseline | | 0.1056 | | 0.1249 | | 0.1976 |
| | Baseline@Spatiotemporal | $\frac{D_v}{D_u}=5$ | 0.0542 | $\frac{D_v}{D_u}=1$ | 0.0698 | $\frac{D_v}{D_u}=100$ | 0.0812 |
| | Ours | | 0.0602 | | 0.0782 | | 0.0853 |
| | Ours@Spatiotemporal | | **0.0469** | | **0.0489** | | **0.0671** |
| Navier-Stokes (2D, FNO) | Baseline | | 0.0487 | | 0.0825 | | 0.0369 |
| | Baseline@Spatiotemporal | $\nu=0.01$ | 0.0442 | $\nu=0.05$ | 0.0743 | $\nu=0.0001$ | 0.0269 |
| | Ours | | **0.0175** | | **0.0222** | | **0.0125** |
| Navier-Stokes (2D, Transformer) | Baseline | | 0.0479 | | 0.0853 | | 0.0685 |
| | Baseline@Spatiotemporal | $\nu=0.01$ | 0.0496 | $\nu=0.05$ | 0.0568 | $\nu=0.0001$ | 0.0402 |
| | Ours | | **0.0265** | | **0.0397** | | **0.0256** |

## D.2 DATA EFFICIENCY AND OUT-OF-DISTRIBUTION GENERALIZATION OF TRANSFORMER FOR 3D NAVIER-STOKES

Similar to what we have studied in Section 4, we aim to also demonstrate three key benefits: data efficiency, long-term physical consistency, and strong generalization in OOD simulations using Transformer, for 3D Navier-Stokes as well.

In Figure 10, we can see that joint training (orange square) on both the original and basic forms of the 3D Navier-Stokes equation consistently reduces normalized RMSE from baseline (blue circle) across varying simulation budgets. This improvement is observed for Transformer architectures, highlighting enhanced data efficiency and generalization, which aligns with the results in Section 4.2.

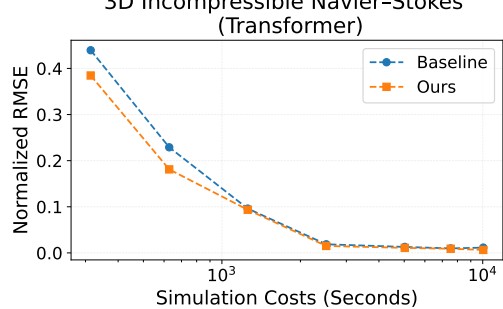

Figure 10: Joint training neural operators on data of the original 3D Navier-Stokes equation and the basic form improves performance and data efficiency.

In Table 6, we show that our joint training approach significantly improves out-of-distribution generalization on 3D Navier-Stokes across all test settings, outperforming the baseline for both FNO and Transformer models. Together with the results in Table 2 and 5, the consistent gains observed across all OOD setting results underscore the effectiveness and robustness of our method in generalizing to previously unseen physical regimes, particularly under significant shifts in simulation parameters.

## D.3 MORE LONG-TERM CONSISTENCY RESULTS OF TRANSFORMER

In our Figure 11, we show the rollout performance of the transformer on the 2D Diffusion-Reaction and 2D incompressible Navier-Stokes equations. Here, we run the experiments with the best checkpoints from training in Figure 9. Losses will be aggregated for five consecutive time steps.

Table 6: Comparisons of OOD generalization on 3D NS for different training methods using Transformer. Models are evaluated using the best checkpoints from training in Figure 10.

| PDE | Model | Source | | Target 1 | | Target 2 | |
|---|---|---|---|---|---|---|---|
| | | Setting | nRMSE | Setting | nRMSE | Setting | nRMSE |
| 3D Navier-Stokes | Baseline | $\nu = 0.01$ | 0.0114 | $\nu = 0.1$ | 0.0327 | $\nu = 0.0001$ | 0.0816 |
| | Ours | | **0.0064** | | **0.0124** | | **0.0322** |

We can see that our improvements in Figure 9 further persist across autoregressive steps, leading to improved long-term consistency, aligning with our results in Figure 6.

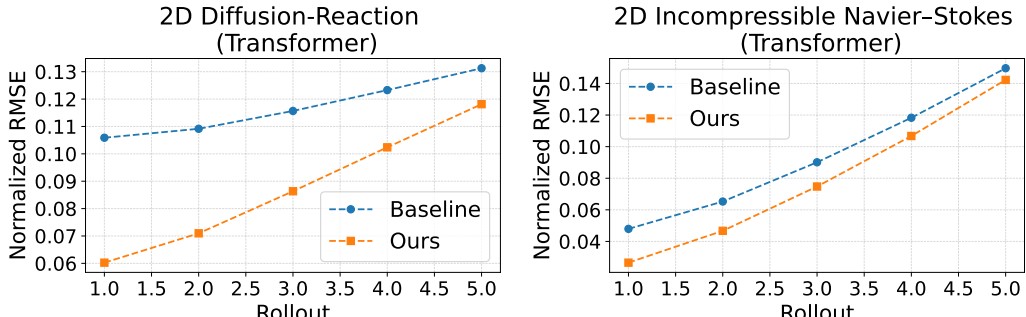

Figure 11: Joint training neural operators on data of the original PDE and the basic form improves performance with autoregressive inference at different unrolled steps using Transformer. Models are evaluated using the best-performing checkpoints from training shown in Figure 9.

### D.4 MORE RANDOM SEEDS

To ensure the statistical robustness of our findings, we now run FNO using three different random seeds during initialization and training. For each configuration, we report the average performance across the three runs, and include standard deviation as error bars in all plots in Figure 12. This enables a more rigorous evaluation of model performance, capturing the inherent variance and mitigating the risk of overinterpretation from single-seed outcomes. We can see that the results demonstrate that joint training of neural operators on data from both the original PDE and its decomposed basic form yields consistent improvements in predictive performance and data efficiency, highlighting the effectiveness of this multiphysics learning strategy.

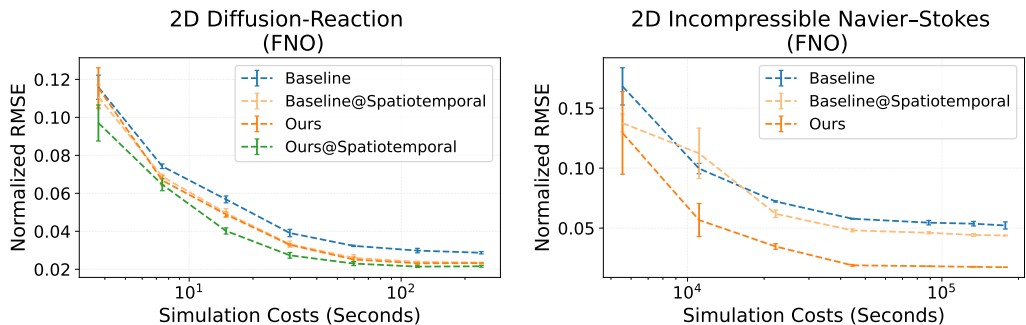

Figure 12: Model performance averaged over three random seeds. Joint training neural operators on data of the original PDE and the basic form consistently improves performance and data efficiency. Dash lines indicate the mean performance, with error bars representing standard deviation. Legends align with the descriptions in Section 4.2. Columns: (left) 2D Diffusion-Reaction, (right) 2D Navier-Stokes. Y-axis: nRMSE. X-axis: Simulation Costs (seconds).

To further demonstrate the reliability and generalizability of our approach, we take the best-performing checkpoints from Figure 12 and evaluate their rollout accuracy as well as their out-of-distribution behavior. From Figure 13 and Table 7, we observe that the standard deviations remain exceptionally

small, underscoring the stability of the model across different random seeds. The results show that joint training of neural operators on both the original PDE and its decomposed basic form yields consistent improvements in long-term physical consistency and generalizes better to unseen physical dynamics. Together, these findings statistically prove that the proposed multiphysics learning strategy leads to more stable, more reliable, and more physically consistent neural operators.

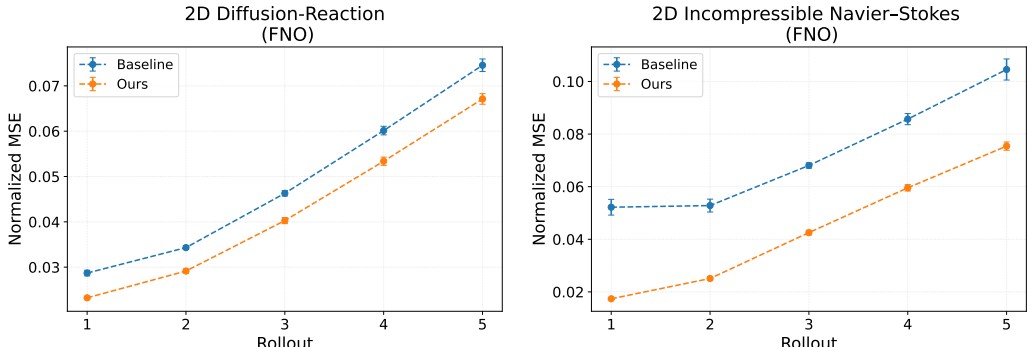

Figure 13: Model performance averaged over three random seeds. Joint training neural operators on data of the original PDE and the basic form consistently improves long-term physical consistency. Dash lines indicate the mean performance, with error bars representing standard deviation. Models are evaluated using the best-performing checkpoints from training shown in Figure 12.

Table 7: Comparisons of OOD generalization for different training methods averaged over three random seeds. Models are evaluated using the best checkpoints from training in Figure 12, under comparable simulation cost settings. "Spatiotemporal": short for "Spatiotemporal Downsampling".

| PDE | Model | Source | | Target 1 | | Target 2 | |
|---|---|---|---|---|---|---|---|
| | | Setting | nRMSE Mean (SD) | Setting | nRMSE Mean (SD) | Setting | nRMSE Mean (SD) |
| Diffusion-Reaction (2D, FNO) | Baseline | $\frac{D_v}{D_u} = 5$ | 0.0287 (0.0006) | $\frac{D_v}{D_u} = 1$ | 0.0411 (0.0008) | $\frac{D_v}{D_u} = 100$ | 0.0754 (0.0014) |
| | Baseline@Spatiotemporal | | 0.0235 (0.0001) | | 0.0305 (0.0005) | | 0.0664 (0.0041) |
| | Ours | | 0.0232 (0.0003) | | 0.0329 (0.0003) | | **0.0532 (0.0014)** |
| | Ours@Spatiotemporal | | **0.0216 (0.0005)** | | **0.0296 (0.0002)** | | 0.0548 (0.0053) |
| Navier-Stokes (2D, FNO) | Baseline | $\nu = 0.01$ | 0.0522 (0.0030) | $\nu = 0.05$ | 0.0855 (0.0028) | $\nu = 0.0001$ | 0.0367 (0.0002) |
| | Baseline@Spatiotemporal | | 0.0462 (0.0017) | | 0.0729 (0.0013) | | 0.0272 (0.0004) |
| | Ours | | **0.0174 (0.0002)** | | **0.0222 (0.0001)** | | **0.0131 (0.0005)** |

## D.5 Loss Reweighting

In Section 3.2, we define our total loss for joint learning of the original PDE ($\text{Loss}_{\text{Full}}$) and its fundamental physical knowledge (decomposed basic form $\text{Loss}_{\text{Basic}}$) as:

$$\text{Loss} = \text{Loss}_{\text{Full}} + 0.7 \times \text{Loss}_{\text{Basic}}.$$

To address the concern regarding the fixed auxiliary loss weight, we conducted an ablation study examining the effect of auxiliary loss weighting in joint training using FNO for the 2D Diffusion–Reaction system. We evaluated three auxiliary weight settings (0.5, 0.7, and 1.0) across the full range of simulation budgets used in the data-efficiency experiments (Figure 5).

As shown in Table 8, the normalized RMSE values are highly consistent across all three weight choices. Importantly, the maximum absolute deviation across weights remains below 0.0063 for all simulation costs, with most gaps around 0.002–0.004. Although performance can be further improved through exhaustive hyperparameter fine-tuning, the observed variations across auxiliary weights are small relative to the overall error scale and substantially smaller than the performance gains achieved by increasing the training budget, which is trivially beyond the scope of our work.

This result demonstrates that the model is largely insensitive to the specific choice of auxiliary weight and suggests that the improvements achieved through joint training are robust with respect to this hyperparameter. Based on this analysis, we fix the auxiliary weight to 0.7 for all the experiments.

Table 8: Ablation study on the auxiliary loss weight in joint training using FNO for the 2D Diffusion-Reaction across three settings: auxiliary weights of 0.5, 0.7, and 1.0. Results indicate that our model is robust to the choice of auxiliary loss weighting. Simulation costs are aligned with the ones used in Figure 5 (top-left).

| Simulation Costs (Seconds) | Weight = 0.5 | Weight = 0.7 | Weight = 1 |
|---|---|---|---|
| 3.73 | 0.1255 | 0.1233 | 0.1214 |
| 7.46 | 0.0674 | 0.0659 | 0.0636 |
| 14.92 | 0.0508 | 0.0478 | 0.0445 |
| 29.84 | 0.0349 | 0.0330 | 0.0314 |
| 59.68 | 0.0274 | 0.0266 | 0.0249 |
| 119.36 | 0.0245 | 0.0238 | 0.0224 |
| 238.72 | 0.0248 | 0.0236 | 0.0221 |

### D.6 CHOICE OF THE FUNDAMENTAL TERM

To demonstrate the importance of the choice of dropping terms, we conduct an ablation study on 2D Diffusion-Reaction and 2D Navier-Stokes. For Diffusion-Reaction, we simulate reaction term instead of the fundamental diffusion term. The simulation cost for reaction-only term is $2.048 \times 10^{-3}$ seconds per step, corresponding to a 1:9 sample mixture ratio when compared to the simulation cost of the original PDE data. For Navier-Stokes, we simulate diffusion term instead of the fundamental advection term. The simulation cost for diffusion-only term is $0.234$ seconds per step, corresponding to 1:12 sample mixture ratio, making sure the results comparable.

From Figure 14, we can find that in 2D Diffusion-Reaction, keeping the reaction term and removing the fundamental diffusion term will damage the accuracy with up to 64% increase of nRMSE compared to the baseline, while applying the fundamental diffusion term keeps boosting the model performance with 11% to 24% decrease of nRMSE. Similarly, keeping diffusion term but dropping advection term in 2D Navier-Stokes can damage the accuracy up to 13% increase of nRMSE compared to baseline and up to 179% increase compared to the one using the functional advection term. This ablation study confirms that **the correct fundamental basic term** can improve the data-efficiency when joint training with the original data, and proved that the source of improvement in Figure 5 is clearly from training with the fundamental term itself.

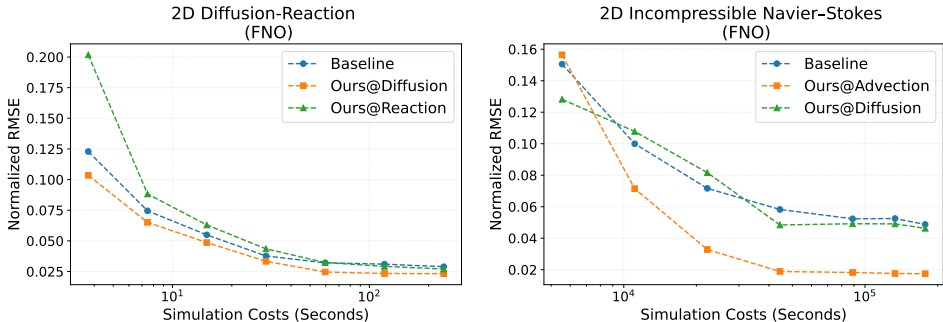

Figure 14: Ablation study of joint training neural operators with two different decomposed terms: the fundamental diffusion term, and reaction term in 2D Diffusion-Reaction, and the fundamental advection term and diffusion term in 2D Navier-Stokes, shows the importance of choice on fundamental terms from PDE equation. Y-axis: nRMSE. X-axis: Simulation Costs (Seconds).

To further clarify that the choice of fundamental term is not ad hoc, we also provide a principled and quantitative criterion for determining which decomposed components should be retained. For each candidate fundamental operator, we compute the RMSE between the full PDE simulation and the corresponding decomposed simulation over the entire simulation trajectory, using identical initial conditions to ensure a fair comparison. This RMSE here serves as a distance metric that quantifies how much each component contributes to the overall dynamics. In 2D Diffusion–Reaction, the RMSE between the full equation and the diffusion-only simulation is 0.553, whereas the RMSE with the reaction-only simulation is much larger at 5.821, which further confirms the results of the qualitative visualization of the original PDE and its two decomposed term we obtained in Figure 15.

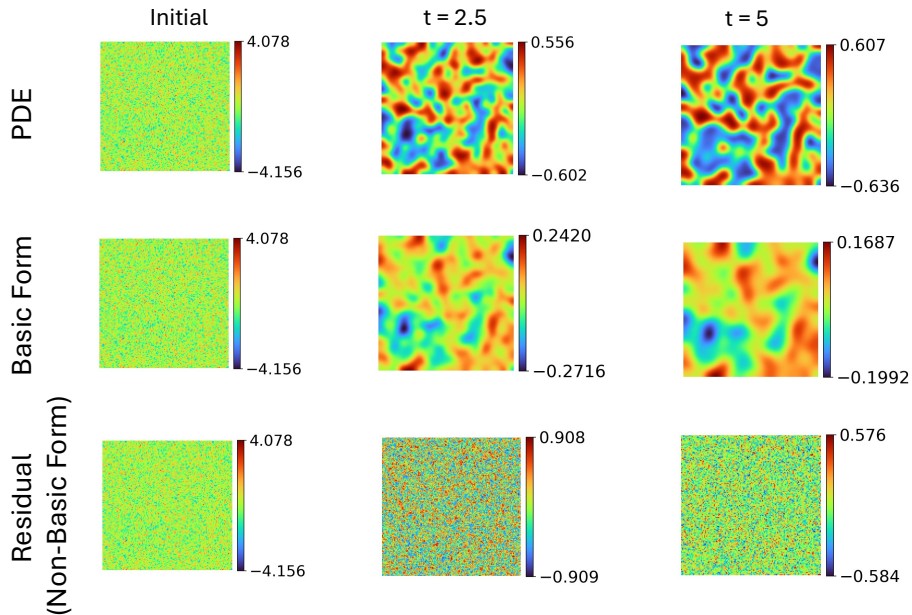

Figure 15: Qualitative visualization of the original PDE and its decomposed forms for 2D Diffusion-Reaction. This results combined with the quantitative results obtained from the RMSE experiment demonstrate that our choice of fundamental term is principled.

## D.7 DPOT (HAO ET AL., 2024) ON 2D DIFFUSION-REACTION

To further demonstrate the scalability of our method to larger models and datasets, we additionally evaluate our method using DPOT (Hao et al., 2024), which is a state-of-the-art foundation model designed specifically for large-scale PDE pretraining. We use the publicly released medium-sized DPOT checkpoint. To adapt our method structure, we applied the model with the task-specific output layers, which contains 124M model parameters. This represents a substantial increase in scale compared to the 0.46M-parameter FNO used in our primary experiments described in Appendix B. We also increase our fine-tuning sample size from 2–128 (See Table 4) to 32–256 for DPOT.

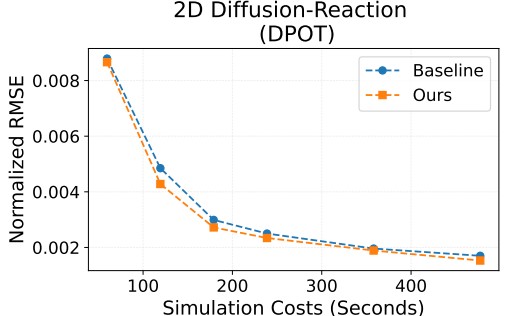

Figure 16: Data efficiency on 2D Diffusion–Reaction using the DPOT foundation model, suggesting joint training on the original and basic form yields consistently lower error across simulation-cost budgets

We then fine-tune DPOT under the same simulation-cost budget for both baseline and our method. Same as the setting in Section 4, we apply our method by reallocating half of the simulation budget to generate auxiliary data from the decomposed basic PDE forms based on the sample mixture ratio defined in Table 1. All experiments use the default DPOT fine-tuning configuration, and apply the same optimization cost (number of gradient descent steps) for both baseline and our method, ensuring a fair and consistent comparison. As shown in Figure 16, our approach continues to improve data efficiency and achieves consistently lower nRMSE compared to fine-tuning DPOT on original PDE data alone across all simulation-cost regimes. Notably, these gains persist despite DPOT's significantly larger capacity and extensive pretraining, demonstrating that our multiphysics auxiliary tasks provide benefits beyond what is already captured by large-scale foundation pretraining.

### D.8 LIE TRANSFORM ARGUMENT ON 2D IMCOMPRESSIBLE NAVIER-STOKES

Lie symmetries offer a way to generate new, physically valid training examples by exploiting the analytic group transformations that map one PDE solution to another. This enables the model to learn representations that are inherently equivariant to fundamental symmetries such as translation, rotation, and scaling. To further prove the strength of our model, we leverage the implementation of Lie point symmetry augmentation from (Brandstetter et al., 2022; Mialon et al., 2023), which is orthogonal to our multiphysics joint training approach, to 2D incompressible Navier Strokes equation.

We incorporate the augmentation process into our model. We only apply Lie-transform augmentations exclusively to the velocity ($u$) of the original 2D incompressible Navier-Stokes, leaving the remaining density and all target variables from the decomposed basic forms unchanged. Following (Mialon et al., 2023), the Lie transformation is implemented with a second-order Lie–Trotter splitting scheme with two steps, where the five fields $(x, y, t, u_x, u_y)$ were transformed in accordance with the sampled generator strengths as follows: a maximum time shift ($g_1$) strength of 0.1, maximum spatial translations ($g_2$, $g_3$) strength of 0.1 in $x$ and $y$ respectively, a maximum scaling ($g_4$) strength of 0.05, a maximum rotation ($g_5$)

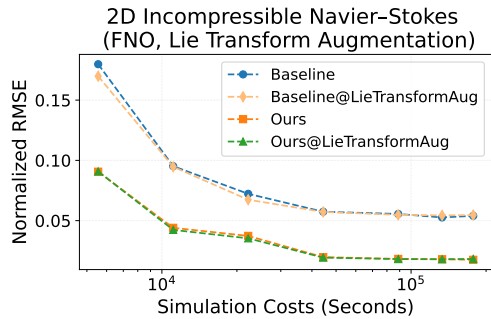

Figure 17: Joint training neural operators on data of the original PDE and the basic form, as a complementary data augmentation orthogonal to Lie-transform augmentation, can further improve performance and data efficiency. Y-axis: normalized RMSE. X-axis: simulation costs (seconds).

strength of $10°$, corresponding to $\pi/18$ radians, a maximum $x$-linear boost ($g_6$) and $y$-linear boost ($g_7$) strength of 0.2 and a maximum $x$- and $y$-quadratic boosts ($g_8$, $g_9$) strength of 0.05.

As our decomposed basic form is orthogonal to the Lie point symmetry augmentation, our method can serve as a complementary data augmentation. In Figure 17, we study prediction errors (nRMSE) of neural operators trained with different numbers of training samples (simulations). As we have already seen (Figure 5), our approach (orange square) significantly outperforms the baseline (blue circle). In contrast, the Lie-transform augmentation alone (yellow diamond) only marginally improves the baseline. As a result, combining our approach with Lie transformations (green triangle) yields strong performance, but is comparable with our approach alone, underscoring the orthogonal and complementary benefits of these two techniques.

### D.9 VISUALIZATION OF PREDICTIONS

To show the predicted PDE solution from our jointly training neural operators on original PDE equation and its basic form aligns with the ground truth, we present qualitative visualizations of model predictions across three PDEs, 2D Diffusion-Reaction, 2D and 3D Incompressible Navier-Stokes, and 1D Kuramoto–Sivashinsky, in Figure 18. For the first three cases, the initial state and predicted states at intermediate and final rollout times are shown. For 1D Kuramoto–Sivashinsky, we show the whole predicted trajectory here. The predictions are generated using the FNO model trained with our joint training framework. Across all systems and time points, the predictions closely align with the expected dynamics, accurately capturing both spatial patterns and temporal evolution. These visualizations highlight the model's capacity to generalize across scales and exhibit physically coherent behavior.

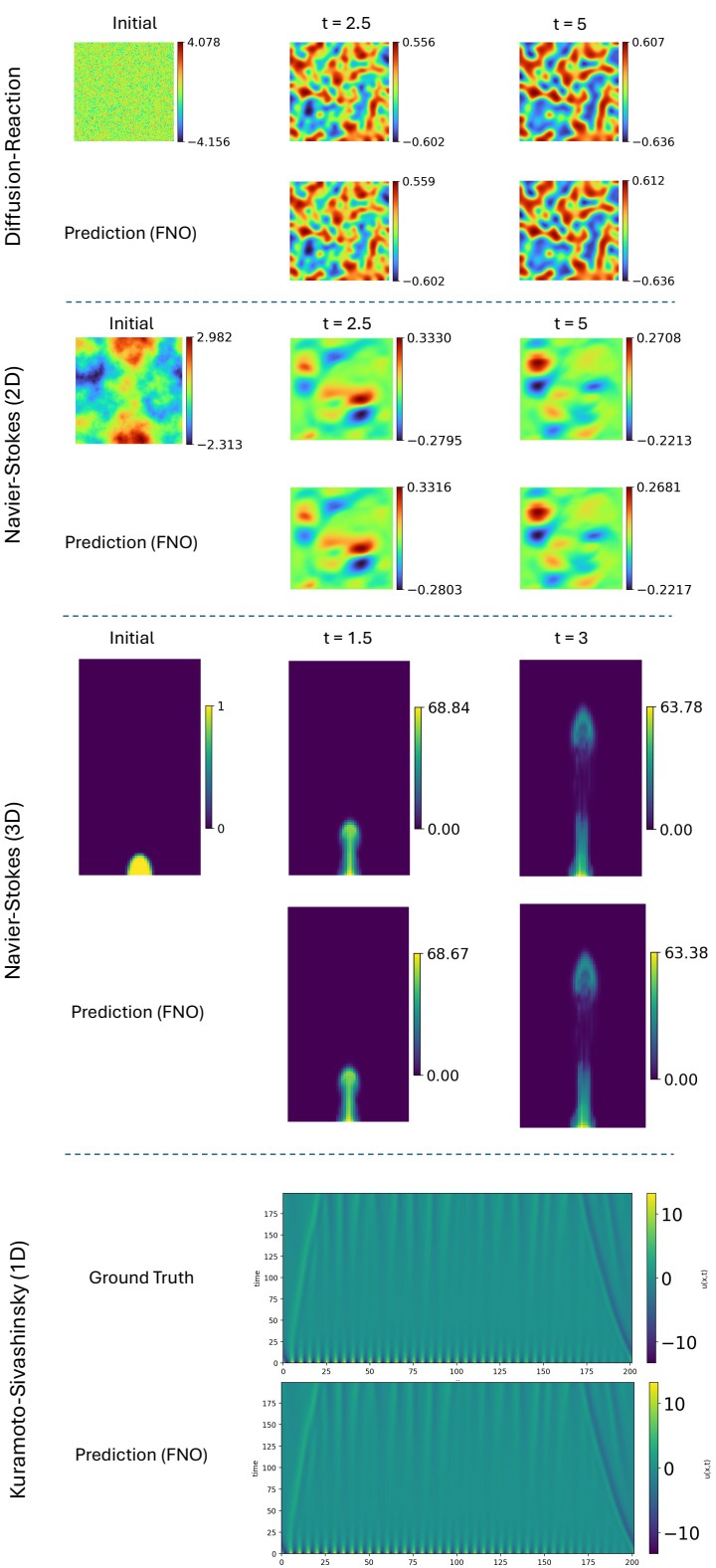

Figure 18: Qualitative visualization of model predictions for 2D Diffusion-Reaction, 2D Incompressible Navier-Stokes, 3D Incompressible Navier-Stokes systems and 1D Kuramoto–Sivashinsky using FNO trained with our joint framework. For the first three cases, the initial state and predicted states at intermediate and final rollout times are shown. For 1D Kuramoto–Sivashinsky, we show the whole predicted trajectory here. The results demonstrate accurate temporal evolution and spatial coherence.

