# OpenReview forum: "Learning Data-Efficient and Generalizable Neural Operators via Fundamental Physics Knowledge"
_ICLR.cc/2026/Conference — ICLR 2026 Poster_

### Official Review · Reviewer_dWHm · 2025-10-27

**Soundness:** 2
**Presentation:** 3
**Contribution:** 2
**Rating:** 4
**Confidence:** 2

**Summary:**

This paper proposes a data-efficient learning framework for PDE dynamics forecasting by jointly learning from both the original PDEs and their simplified basic forms. Extensive experiments on a wide range of 1D/2D/3D PDE problems demonstrates the effectiveness of the proposed framework.

**Strengths:**

- The proposed method is well-motivated. The authors provide a critical observation by evaluating existing SciML foundation models. They find a strong correlation between a model's performance on the original PDE and its performance on the fundamental components of that PDE (e.g., pure diffusion for a reaction-diffusion system). However, the absolute error on these basic terms remains high, indicating that even powerful models lack a robust understanding of the foundational physics, which motivates the need for explicit training on these concepts.
- Methodological Innovation:​​ The paper proposes a simple yet effective multiphysics training framework. It first derive a "basic form" from the original PDE by retaining terms governing essential dynamics and removing terms that cause computational stiffness or high cost. The model is trained on a composite dataset from simulations of both the original PDE and the basic form.

**Weaknesses:**

- Heuristic Nature of Decomposition:​​ The process for selecting terms for the "basic form," while physically intuitive, remains heuristic. A more formalized principle or an ablation study discussing the impact of alternative decompositions more prominently would strengthen the methodology.
- Inadequate Mechanistic Explanation for the Efficacy of Basic Form Data​: A significant weakness of the paper lies in its insufficient exploration of the underlying mechanisms by which the "basic form" data aids the learning of the original PDE. The attribution of performance gains solely to the incorporation of "fundamental physics knowledge" is a high-level concept that lacks granularity. A more rigorous analysis is required to dissect how the basic form data contributes. A possible explanation is that data from the basic form may provide more diverse initial conditions. Can the data from the basic form be replaced with an equivalent amount of original PDE data? Although this would incur greater simulation costs, it would help clarify the specific ways in which data from the basic form aids the model in learning the original PDE.

**Questions:**

See weaknesses.

---

> ### Author Response · Authors · 2025-11-21
>
> First, we would like to sincerely thank Reviewer dWHm for the time and effort dedicated to reviewing our paper and highlighting the motivation and innovation of our method! We truly appreciate the feedback and the assessment of our work.
>
>
> > **Q1** Heuristic Nature of Decomposition
>
> Thank you for your question. Rigorously identifying the best basic term is essential. To better demonstrate our method, we have the following ablation studies that *swap* the choice of the basic term: 1) 2D Diffusion-Reaction (diffusion was the basic term in our paper, now we simulate the **reaction term**; see Appendix D.6); 2) we further added an ablation study in 2D Navier-Stokes, (advection was the basic term in our paper; now we simulate the **diffusion term**). We have updated the results in the revised version. Please kindly check our latest PDF submission. Note that we always keep the total simulation cost the same across all experiments for one PDE. Results: 1) in 2D Diffusion-Reaction, keeping the reaction term and removing the fundamental diffusion term will damage the accuracy **with up to 64% increase of nRMSE** compared to baseline; 2) in 2D Navier-Stokes, keeping diffusion term while removing advection term will damage the accuracy **with up to 13% increase of nRMSE** to baseline and **up to 1.79 times of nRMSE** to the one obtained from the fundamental advection term. These two ablation studies confirm that it is the joint training with the fundamental basic PDE term that leads to the improvements.
>
>
>
> > **Q2** Intuition of the efficacy of basic form data
>
> Thank you for the question!
> Here we highlight our core mechanistic explanation and motivations:
> * Lines 48–50: Unlike numerical solvers, neural operators lack rigorous verification.
> * Section 2.2: Neural operators appear to achieve low PDE approximation errors, but they do not truly “understand” PDEs, as they incur much higher errors when we retain only the fundamental governing PDE with the non-dominant / non-leading terms removed.
>     * In Figure 4, we show that the visualizations of the simulations of the basic and original PDEs are highly relevant and aligned. This provides qualitative evidence that the basic PDE form is not just a naive simplification, while it generates physically plausible trajectories that resemble the original PDE.
>
> Based on these intuitions, here is how the basic form data contributes: Our approach explicitly enforces neural operators to learn both the original governing equation and its fundamental terms.
>
> Our response to your previous question further adds granularity on how to properly include contributions from basic PDE forms: our ablation studies show that *swapping* the basic term consistently harms performance, **even under the same simulation-cost budget**. Replacing fundamental diffusion with the reaction term in 2D Diffusion–Reaction increases nRMSE by up to **64%** compared to baseline, and replacing fundamental advection with diffusion in 2D Navier–Stokes leads to up to **13%** higher error than baseline and **1.79 times** the error of using the correct basic term. These results confirm that improvements arise specifically from incorporating the *fundamental basic* PDE term, rather than from including arbitrary simplified components.
>
> We promise to make these explanations clearer in our camera-ready version.
>
>
> > **Q3**  Regarding if we can replace with an equivalent amount of original PDE data
>
> Thank you for the question!
> We would like to emphasize the importance of keeping all comparisons in the paper performed under equal simulation cost to ensure fairness across methods. Replacing the basic-form data with an equivalent amount of original PDE data would require substantially higher simulation cost, because simulating the full PDE is significantly more expensive than simulating its simplified components. Such a comparison would therefore violate the equal-cost constraint and confound the outcome we seek to measure.
>
> Under the same simulation budget, however, we find that joint training with the basic forms provides greater performance gains than using only original PDE data. In addition, this question can be further clarified by examining the baseline curves in Figure 5: In 2D Navier–Stokes (FNO; top-right panel), even when the simulation-cost budget increases and more full-PDE samples are added, the baseline shows **only modest**  gains: the nRMSE decreases from 0.0582 to 0.0487 when the primary training samples increase from 16 to 64. In contrast, our method already achieves 0.0328 using combined samples with both original PDE data and its basic form generated with a cost equivalent to only 8 primary simulations, and ultimately reaches 0.0174 under the same overall simulation budget, substantially outperforming the baseline at every cost level. This highlights the ability of our method using the physics-inspired basic forms to provide a more data-efficient improvement.

---

> > ### Comment · Reviewer_dWHm · 2025-11-27
> >
> > I thank the authors for their detailed and clear responses. Most of my concerns have been addressed and I have raised my score to 6.

---

> ### Author Response · Authors · 2025-11-24
> **Looking forward to more discussions.**
>
> Dear Reviewer dWHm,
>
> We would like to kindly remind you that the author-reviewer discussion period has started for several days now.
>
> We would greatly appreciate it if you could review our responses to your initial comments at your earliest convenience.
>
> This will enable us to address any additional queries or feedback you might have before the discussion period ends.
>
> Should our responses sufficiently address your concerns, we respectfully request that you consider raising the rating of our work.
>
> Thank you very much for your attention, time, and efforts!
>
> Best regards,
>
> Authors of Submission 14708

---

### Official Review · Reviewer_kRiv · 2025-10-30

**Soundness:** 2
**Presentation:** 3
**Contribution:** 2
**Rating:** 4
**Confidence:** 5

**Summary:**

The paper proposes a multiphysics training scheme that jointly learns from full PDE simulations and their decomposed “basic forms” the authors argue that this injects “fundamental physics knowledge” into neural operators (NOs) and improves data efficiency and OOD generalization.  The key contribution lies in identifying and leveraging "fundamental physics knowledge" through decomposed basic PDE forms.  This has not  been explored extensively in the neural operator literature.
The authors target two central SciML issues, 1. data hunger and 2. poor OOD transfer, for operator learning across 1D/2D/3D PDEs (Diffusion-Reaction, Navier–Stokes, Kuramoto–Sivashinsky, plus ScalarFlow).
Formulations of PDEs and “basic forms” are standard and correctly specified
The paper is generally well written with helpful overview figures (Fig. 3 pipeline; Fig. 4 gallery of PDEs/basic forms) and plots tying simulation cost to nRMSE. Implementation, data splits, and training schedules are placed in appendices.
Minor typos remain but do not impede readability.
Central claims are supported with experiments and results that are a bit light on content
The validation of physics (central theme) is light given no explicit checks on mass/energy conservations.  Another issue is the heuristic treatment of the fundamental physics term.

**Strengths:**

The authors target two central SciML issues, 1. data hunger and 2. poor OOD transfer, for operator learning across 1D/2D/3D PDEs (Diffusion-Reaction, Navier–Stokes, Kuramoto–Sivashinsky, plus ScalarFlow).
The key contribution lies in identifying and leveraging "fundamental physics knowledge" through decomposed basic PDE forms.  This has not  been explored extensively in the neural operator literature.
Proposed benefits such as:
Data efficiency, Long horizon stability , OOD generalization , are all desirable.

**Weaknesses:**

The term "fundamental physics knowledge" is somewhat vague and could be better defined
Section 3.1 could be more systematic in explaining the decomposition principles
Some notation inconsistencies (e.g., switching between v and u for solutions

Missing error bars in main results (added later in appendix)
there is limited statistical analysis
The ScalarFlow experiment (Section 4.5) is somewhat disconnected and brief
Claims about "fundamental physics knowledge" being key are not fully validated (could be just multi-task regularization)
There is a lack of theoretical insight, no formal explanation of why the approach works beyond intuition is provided.
The decomposition rules in Section 3.1.1 seem ad-hoc without principled justification

Evaluation is very basic, results only compare against vanilla baseline and spatiotemporal downsampling
There is no comparison/discussion with other data-efficient methods or recent foundation models
Real-world evaluation is very limited (only ScalarFlow)
Inconsistent terminology: "Fundamental physics knowledge" vs "basic forms" used interchangeably
Missing details: How are the mixture ratios exactly determined? Training time comparisons?
Limited discussion: When would this approach fail? What about PDEs that don't decompose nicely?
Presentation issues: Some figures (especially in appendix) are too small to read clearly
Scalability concerns: All experiments on relatively small-scale problems

Grammer-Typos
line 483: and outha ha h-of-distribution generalization.

**Questions:**

See weakness section , addressing those would be good
recommend to:
Better justify the decomposition principles
Provide theoretical analysis or at least intuition for why the approach works
Compare with more baselines
Discuss limitations and failure cases more thoroughly

**Details Of Ethics Concerns:**

no ethical issues

---

> ### Author Response · Authors · 2025-11-21
>
> First, we sincerely thank Reviewer kRiv for the time and effort dedicated to reviewing our paper and highlighting the novelty of our work.
>
> > **Q1** Intuition for why the approach works
>
> Thank you for the question!
> Here we highlight our core intuitions and motivations:
> * Lines 48–50: Unlike numerical solvers, neural operators lack rigorous verification.
> * Section 2.2: Neural operators appear to achieve low PDE approximation errors, but they do not truly “understand” PDEs, as they incur much higher errors when we retain only the fundamental governing PDE with the non-dominant / non-leading terms removed.
>     * In Figure 4, we show that the visualizations of the simulations of the basic and original PDEs are highly relevant and aligned. This provides qualitative evidence that the basic PDE form is not just a naive simplification, while it generates physically plausible trajectories that resemble the original PDE.
>
> Based on these intuitions, here is why our approach works: Our approach explicitly enforces neural operators to learn both the original governing equation and its fundamental terms.
>
> We define our “fundamental physics knowledge” in Lines 150–156. In Sections 3.1.2–3.1.4, for each equation, we systematically provide detailed explanations of the rationale behind our choice of fundamental terms.
> Here, we reiterate our definition:
> 1. We keep the terms that retain the essential and dominant dynamics, such as diffusion in Diffusion-Reaction and advection in incompressible Navier-Stokes.
> 2. We drop the terms that increase computational cost while contributing less to the pattern formation of interest.
>
> We promise to make this definition clearer in our camera-ready version.
>
>
> > **Q2**  Scenarios of approach failing or PDEs not decomposing nicely
>
> Thank you for the question! Following the first part, this approach will fail if we do not define the fundamental terms correctly.
>
> Rigorously identifying the best basic term is essential. To better demonstrate our method, we have the following ablation studies that *swap* the choice of the basic term: 1) 2D Diffusion-Reaction (diffusion was the basic term in our paper, now we simulate the **reaction term**; see Appendix D.6); 2) we further added an ablation study in 2D Navier-Stokes, (advection was the basic term in our paper; now we simulate the **diffusion term**). We keep the simulation cost the same across all experiments for one PDE. Results: 1) in 2D Diffusion-Reaction, keeping the reaction and removing the fundamental diffusion will damage the accuracy **with up to 64% increase of nRMSE** compared to baseline; 2) in 2D Navier-Stokes, keeping diffusion while removing advection will damage the accuracy **with up to 13% increase of nRMSE** to baseline and **up to 1.79 times of nRMSE** to the one obtained by advection term. These two ablation studies confirm that it is the **joint training with the fundamental PDE term** that leads to the improvements, and failing to recognize the term that contributes most to the pattern will result in the failure to improvements.
>
> For the second part: if the governing equation is fundamental (for example, the Euler equation that does not have the viscosity term), we cannot decompose any further. Please kindly let us know if you would like to discuss other PDE cases in your mind.
>
>
> > **Q3** Limited statistical analysis
>
> Thank you for pointing this out. In Appendix D.4, we provide a statistical analysis using three independent random seeds, where we report mean performance and standard deviations for all configurations for data-efficiency in 2D Diffusion-Reaction and 2D Navier Stokes. These results confirm that our improvements are consistent across seeds, further validating the robustness of our multiphysics learning strategy. In the revised version, we have updated Appendix D.4 with further **adding error bars (standard deviation)** covering **rollout accuracy and OOD generalization**. Please kindly check our latest PDF submission. These additional results confirm that the improvements reported in the main paper are consistent and robust across seeds.
>
> > **Q4** Disconnection of the ScalarFlow
>
> Thank you. We include the ScalarFlow experiment to demonstrate a real-world application of our method and to show how our physics-inspired auxiliary training improves robustness in the complex real-world scenarios beyond controlled numerical benchmarks. We would like to clarify that this experiment is **not** meant as a standalone numerical benchmark, but as a validation that our method helps **bridge the domain gap between numerical simulations and real-world measurements**. By pretraining on 3D Navier-Stokes simulations and further evaluating on ScalarFlow, we show that the proposed strategy continues to provide gains even under the high complexity and variability of real smoke data. This supports our broader claim that incorporating fundamental physics structure enhances transferability and generalizability in practical scenarios.

---

> ### Author Response · Authors · 2025-11-21
>
> > **Q5** Concerns regarding scalability and evaluation on other data-efficient methods and recent foundation models.
>
> Thank you for the question! To further address the reviewer’s concerns regarding evaluations on recent foundation-model approaches and scalability to larger models and datasets, we additionally evaluate our method using DPOT [1]. We consider the medium-sized DPOT model of 124M parameters. We have updated this new result in Appendix D.7 (We put the result table below and for more details please kindly check our latest PDF submission!). We found that across all simulation-cost regimes, our method achieves consistently lower normalized RMSE compared to fine-tuning DPOT on original PDE data alone, with the same simulation budgets and optimization costs. As what reviewers dWHm and FhDJ mentioned, our method is **simple yet effective** and **shows consistent improvements**. This experiment reinforces two key points raised by Reviewer kRiv: (1) Our method shows consistent improvement despite DPOT’s significantly larger capacity and extensive pretraining, demonstrating that our auxiliary tasks provide benefits beyond what is already captured by large-scale foundation pretraining; and (2) it scales successfully to large-parameter models. Together, these results strengthen the generalizability and practical applicability of our approach beyond the small-scale settings in the main experiments.
>
> | Sample Size | Simulation Costs (Seconds) | Baseline (DPOT) | Ours (DPOT) |
> | ----------- | -------------------------- | --------------- | ----------- |
> | ds32        | 59.68                      | 0.00879         | 0.00866     |
> | ds64        | 119.36                     | 0.00485         | 0.00428     |
> | ds96        | 179.04                     | 0.00299         | 0.00272     |
> | ds128       | 238.72                     | 0.00250          | 0.00234     |
> | ds192       | 358.08                     | 0.00196         | 0.00189     |
> | ds256       | 477.44                     | 0.00170          | 0.00153     |
>
>
> Also, we would like to further address the concern regarding limited comparisons with other data-efficient or augmentation strategies. In Appendix D.8, we evaluated our method alongside Lie transform data augmentation [2]. As shown in Figure 16, Lie-transform augmentation alone yields only marginal improvements over the baseline, while our method continues to produce **substantial improvements** across all simulation-cost levels, and further combining Lie transforms with our method provides additional benefits. This result further demonstrates the effectiveness and generalizability of our method and proves that the improvements of our approach are not simply due to general data-augmentation effects, but arise specifically from leveraging the physics-inspired decomposition underlying the basic PDE forms.
>
> [1] Hao et al. 2024. Dpot: Auto-regressive denoising operator transformer for large-scale pde pre-training.
>
> [2] Brandstetter et al. 2022. Lie point symmetry data augmentation for neural pde solvers. In International Conference on Machine Learning, pages 2241–2256.
>
>
> > **Q6** Details on sample mixture ratios and training time comparisons
>
> Thanks for the question! We appreciate the review’s attention to the definition of the sample mixture ratio. In Table 1, we report the simulation cost of each PDE and their decomposed term. The sample mixture ratio is **not empirically determined**, but derived in principle: it is based on the rate of the simulation cost of the original PDE and its corresponding fundamental basic term, which ensures **a comparable or even reduced simulation budget** when training with our proposed method.
> To further the fair comparison of the simulation cost, we list the simulation details in Appendix A with respect to the numerical solver,  the choice of grid size, time step, initial and boundary conditions. These settings are fixed throughout the experiments to ensure numerical stability of solvers. Moreover, all simulation cost measurements were conducted under the same CPU/GPU resources. We conducted our experiments on NVIDIA RTX 6000 Ada GPUs, each with 48 GB of memory. This further ensures that the differences in reported cost are solely from the PDE complexity.
>
> Regarding the training time comparison, we list all the training details in Table 4. We want to emphasize that to fairly compare with the baseline, we adopt the same hyperparameters and optimization costs (number of gradient descent steps).
>
>
> > **Q7** Typos and notations
>
> Thank you for pointing these out! We have fixed these in the updated version and will further improve in the camera ready version.

---

> ### Author Response · Authors · 2025-11-24
> **Looking forward to more discussions**
>
> Dear Reviewer kRiv,
>
> We would like to kindly remind you that the author-reviewer discussion period has started for several days now.
>
> We would greatly appreciate it if you could review our responses to your initial comments at your earliest convenience.
>
> This will enable us to address any additional queries or feedback you might have before the discussion period ends.
>
> Should our responses sufficiently address your concerns, we respectfully request that you consider raising the rating of our work.
>
> Thank you very much for your attention, time, and efforts!
>
> Best regards,
>
> Authors of Submission 14708

---

> ### Comment · Reviewer_kRiv · 2025-11-26
>
> The question on using “fundamental physics knowledge" was more on identifying these fundamental knowledge structures in the class of problems where the complete set of underlying physics may not be known or known only partially.     The usage for the standard systems, eg fluid, heat problems etc. is quite obvious.
> Ablation studies on diffusions, advection terms etc are obvious choices, what i was looking for was a formal method regarding these choices, can these fundamental choices be made discoverable ?  For example in complex heterogeneous catalysis with adsorption over multiple catalyst sites  (some of which are in the deeper layers), other than surface adsorption, how does one decide what other adsorption sites to consider as fundamental ?
> In this regard I am struggling to see the general usage outside of well known PDE structures.

---

> ### Author Response · Authors · 2025-11-27
>
> We sincerely thank the reviewer for this insightful comment. To further clarify that the choice of auxiliary physics is not ad-hoc, we can provide a principled and quantitative metric for explaining which decomposed components should be retained (i.e., to make the fundamental choices more discoverable): For each candidate fundamental operator (e.g., diffusion, reaction), we compute the **RMSE between the full PDE simulation and the corresponding decomposed simulation over the entire simulation trajectory, using identical initial conditions to ensure a fair comparison**. This RMSE serves as a distance metric that quantifies if a component dominates the overall dynamics.
>
> For example, in the Diffusion–Reaction system, the RMSE between the full equation and the diffusion-only simulation is **0.553**, whereas the RMSE with the reaction-only simulation is much larger at **5.821**, further confirming the qualitative differences illustrated in **Figure 4**, and also **Figure 15**, a new plot we added to show the qualitative visualization of the original PDE, its basic term, and the residual (i.e., the non-basic form) for 2D Diffusion-Reaction. Please kindly check our latest PDF submission. This RMSE-based criterion provides a systematic way to evaluate the relevance of candidate auxiliary operators, thereby addressing the reviewer’s request for a principled method to identify “fundamental physics” even when deeper or more complex underlying mechanisms are only partially known.

---

### Official Review · Reviewer_FhDJ · 2025-10-30

**Soundness:** 4
**Presentation:** 3
**Contribution:** 3
**Rating:** 6
**Confidence:** 4

**Summary:**

The paper discusses basic forms of standard PDEs. It finds out that if put half of the data generation efforts into generating basic form of PDE, the performance and generalization will generally improves. The paper test on diffusion-reaction, 2d NS, 3d NS, and KS equation. The improvement on 2D NS is the most significant.

**Strengths:**

- The paper studies an important but often overlooked probably on efficient data generation for neural operators.
- When comparing against the baseline, the paper controls the overall run time budget.
- The experiment show consistent improvement.

**Weaknesses:**

- It is not always obviously which is the best basic form to each target equation. I don't think there is a canonical choice, and the performance depends on the choise of augmented basic PDE. For example on 2D NS the treatment is more significant, but not as much on KS. It would be better to add some ablations to justify the choice.
- Overall I like this paper but I don't like the storytelling. The main message should be "it is helpful to generate additional data in simpler form". It is a bit speculative to claim about "fundamental physics knowledge". In my opinion it can be awkward to say diffusion is the "fundamental physics" to diffusion reaction equation, or convection is "fundamental physics" to Navier Stokes equation.
- Line 108, the authors use Spearman correlation which is defined for ordinal (rank) correlation. Instead it would be better to use Pearson correlation as the rank does not matter here.

**Questions:**

- In the experiment, we need to be a bit careful how we measure the simulation cost (Table 1). In practice, the runtime of numerical solvers depends on the choices of gridsize, timestep, and numerical tolerance etc. Here how the simulation cost is measured?
- If we instead add low fidelity data (with lower gridsize) with the same equation, would the performance improve?
- How about adding smaller RE on NS, it would requires low simulation cost too?
In general it would be helpful to add a bit more discussion on the simulation cost.

---

> ### Author Response · Authors · 2025-11-21
>
> First, we would like to sincerely thank Reviewer FhDJ for the time and effort dedicated to reviewing our paper and highlighting the significance of the problem our paper attempts to study. We truly appreciate the feedback and the positive assessment of our work, which has been very encouraging to us.
>
> > **Q1** Choice of the best basic form and the reason for claiming "fundamental physics knowledge"
>
>  Thank you for the question.
> In this research, the best basic term is **designed in principle**, as was discussed in the Line 199-213， 230-250 and 288-304. The key principles of selecting the dropping term are:
> 1. Keep the terms that can retain the essential and dominant physics dynamics, such as diffusion in Diffusion-Reaction and advection in Incompressible Navier-Stokes;
> 2. Drop the terms that will increase the computational cost as well as contribute less to the pattern formation of interest.
>
> Rigorously identifying the best basic term is essential. To better demonstrate our method, we have the following ablation studies that *swap* the choice of the basic term: 1) 2D Diffusion-Reaction (diffusion was the basic term in our paper, now we simulate the **reaction term**; see Appendix D.6); 2) we further added an ablation study in 2D Navier-Stokes, (advection was the basic term in our paper; now we simulate the **diffusion term**). We always keep the total simulation cost the same across all experiments for one PDE. Results: 1) in 2D Diffusion-Reaction, keeping the reaction term and removing the fundamental diffusion term will damage the accuracy **with up to 64% increase of nRMSE** compared to baseline; 2) in 2D Navier-Stokes, keeping diffusion term while removing advection term will damage the accuracy **with up to 13% increase of nRMSE** to baseline and **up to 1.79 times of nRMSE** to the one obtained from the fundamental advection term. These two ablation studies confirm that it is the joint training with the fundamental basic PDE term that leads to the improvements.
>
>
> > **Q2** Correction to Pearson correlation
>
> Thank you for pointing this out! We have applied this in the revised version. The Pearson correlation between errors on original PDEs and their basic terms is 0.9625.
>
> > **Q3**  Simulation cost measurement
>
> Thanks for the question! We appreciate the review’s attention to the definition of our simulation cost. In our paper, we list the simulation details in Appendix A with respect to the numerical solver,  the choice of grid size, time step, initial conditions and boundaries. These settings are **fixed** throughout the experiments to ensure the numerical stability of solvers. Moreover, all simulation cost measurements were conducted under the same CPU/GPU resources. We conducted our experiments on NVIDIA RTX 6000 Ada GPUs, each with 48 GB of memory. This further ensures that the differences in reported cost are **solely** from the PDE complexity.
>
>
> > **Q4**  Model performance by adding low-fidelity data
>
> Thanks for the question! Yes, low-fidelity data with a lower grid size is a common strategy for reducing simulation cost. This corresponds exactly to the *Baseline@Spatiotemporal Downsampling* in our study in Appendix D.1. However, using low-fidelity data does **not introduce new physics knowledge** as the governing equation remains the same. We showed in Figure 9 that our approach, by introducing physically valid auxiliary data generated from basic PDE forms,  enables better data efficiency and stronger out-of-distribution generalization. In fact, this low-fidelity data is orthogonal and complementary to our method. As shown in our experiments (green triangle *Ours@Spatiotemporal* vs. yellow diamond *Baseline@Spatiotemporal* in Fig. 9), combining both, i.e., simulating the basic PDE term at low spatiotemporal resolution, can achieve lower simulation cost and better performance than using low-fidelity data alone.
>
> > **Q5**  Model performance by using a smaller Reynolds number on Navier-Stokes
>
> Thank you for your question! Using a smaller Reynolds number (Re) will reduce the simulation cost when all other numerical settings remain fixed. In our out-of-distribution experiment, we observe that increasing the viscosity from 0.01 to 0.05 (i.e., reducing the Reynolds number by five times) will only marginally reduce the simulation cost from 2.775 secs per step to 2.745 secs per step with respect to the simulation setting we described in Appendix A. However, our study focuses on understanding whether auxiliary decomposed PDEs can improve model performances beyond what can be achieved by modifying physical parameters of the same equation. Even when evaluated at the same Re, the decomposed basic PDEs consistently provide additional benefits, suggesting that the gains arise from the proposed multiphysics learning framework rather than simply from making the primary PDE easier to simulate.

---

> ### Author Response · Authors · 2025-11-24
> **Looking forward to more discussions**
>
> Dear Reviewer FhDJ,
>
> We would like to kindly remind you that the author-reviewer discussion period has started for several days now.
>
> We would greatly appreciate it if you could review our responses to your initial comments at your earliest convenience.
>
> This will enable us to address any additional queries or feedback you might have before the discussion period ends.
>
> Should our responses sufficiently address your concerns, we respectfully request that you consider raising the rating of our work.
>
> Thank you very much for your attention, time, and efforts!
>
> Best regards,
>
> Authors of Submission 14708

---

> > ### Comment · Reviewer_FhDJ · 2025-11-27
> >
> > Dear Authors,
> >
> > Thank you for the clarification and the additional experiments. These updates partially address my questions and concerns. I appreciate the paper and am leaning toward acceptance, but the improvements are not sufficient for me to raise my score. The current score (6) reflects my objective and honest assessment.
> >
> > It is well understood that pre-training on related datasets can be beneficial (as demonstrated, for example, in MPP, DPOT, and other works discussed in the paper). While the paper frames its dataset construction as leveraging “fundamental physics knowledge,” the current results do not yet convincingly demonstrate such fundamental understanding of dataset construction. It would be valuable to develop a more systematic approach to data generation that better substantiates this claim.
> >
> > Thank you,
> > Reviewer

---

### Author Response · Authors · 2025-11-30

Dear PC, AC, and reviewers:

Since further public discussion is no longer permitted, we would like to provide a rebuttal summary here to document all revisions and updates made during the rebuttal period. We sincerely thank all reviewers for their thoughtful and constructive feedback, which has helped improve the quality and clarity of our work.

Below, we summarize the main questions raised by the reviewers and describe how we have thoroughly addressed each of them. Notably, prior to the reversion, Reviewer dWHm expressed satisfaction with our detailed responses and increased the score from 4 to 6 (please refer to the reviewer’s latest comment). Accordingly, before the revert, our final scores were:
- Reviewer FhDJ: 6
- Reviewer dWHm: 6
- Reviewer kRiv: 4

> **Core Contribution: Principled and Intuitive PDE Decomposition for Learning Data-Efficient and Generalizable Neural Operators**

>> Nature of Decomposition (Choice of basic form) ([FhDJ Q1](https://openreview.net/forum?id=mJiPqOzc3O&noteId=ZPmGLsqpnJ), [dWHm Q1](https://openreview.net/forum?id=mJiPqOzc3O&noteId=9i14rr7sNp))

In this work, the choice of the basic term is principled rather than ad-hoc, as detailed in Lines 199–213, 230–250, and 288–304. We retain only the terms that capture the essential, dominant physical dynamics and drop those that add computational cost while contributing little to the patterns of interest.

We conduct additional ablation studies demonstrating that *using the correct fundamental term is essential*. Building on our Diffusion-Reaction ablation with a reaction-only term (Appendix D.6), we further replace advection with diffusion in 2D Navier–Stokes, which results in up to **13% higher** error than baseline and **1.79 times** the error obtained when using the correct basic term.

>> Intuition ([kRiv Q1](https://openreview.net/forum?id=mJiPqOzc3O&noteId=mcVaWUSybA), [dWHm Q2](https://openreview.net/forum?id=mJiPqOzc3O&noteId=9i14rr7sNp))

Our intuition is grounded in two observations: 1) neural operators, unlike numerical solvers, *lack rigorous verification* (Lines 48–50); and 2) although they achieve low approximation error, they do *NOT* truly “understand” PDEs as they incur much higher errors when only the fundamental PDE terms are retained (Section 2.2). Figure 4 further shows qualitative evidence that the basic PDE form produces physically plausible trajectories closely aligned with the full equation, indicating that it captures the essential dynamics rather than being a naive simplification.

Motivated by this, our method *explicitly* enforces neural operators to learn both the full governing equation and its fundamental terms. As further supported by our ablations above, replacing the true fundamental term with an incorrect one consistently degrades performance, confirming that the improvements are from incorporating the correct underlying physics rather than arbitrary simplifications.


>> [One further principled, intuitive, and quantitative metric for PDE decomposition](https://openreview.net/forum?id=mJiPqOzc3O&noteId=JWesL3I2nq)

We introduce a RMSE-based metric to *quantitatively identify* dominant physics in a PDE, even when the full governing equation is only partially known. For each candidate basic term, we compute the RMSE between full-PDE simulated trajectory and each candidate decomposed simulated trajectory under identical initial conditions. This directly measures each term’s contribution to the true dynamics: For example, in Diffusion–Reaction, diffusion-only yields an RMSE of 0.553, whereas reaction-only produces 5.821, which is consistent with the qualitative alignment in Figures 4 and 15. This metric thereby provides a principled and reproducible way to guide PDE decomposition.


> **Scale Up to SciML Foundation Models**

To demonstrate scalability, we add new experiments using the 124M-parameter DPOT SciML foundation model. As shown in Appendix D.7, our method consistently achieves lower normalized RMSE than fine-tuning DPOT on the original PDE data alone across all simulation-cost regimes, despite DPOT’s extensive pretraining. This confirms that our physics-inspired auxiliary tasks provide complementary benefits beyond what large-scale models already capture.

We also evaluated the Lie-transform augmentation baseline (Appendix D.8). While Lie-transform baseline yields only marginal improvements, our method delivers substantial and consistent gains, with additional benefits when combined. These results show that the performance boost comes from our physics-driven decomposition instead of general data-augmentation effects, and our approach scales effectively to large SciML foundation models.

---

> ### Author Response · Authors · 2025-11-30
>
> > **Additional explanations or justifications:**
>
> - Improved explanations of simulation setting and sample mixture ratio: [FhDJ Q3](https://openreview.net/forum?id=mJiPqOzc3O&noteId=ZPmGLsqpnJ), [kRiv Q6](https://openreview.net/forum?id=mJiPqOzc3O&noteId=g9FkVZ5F8p) and [dWHm Q3](https://openreview.net/forum?id=mJiPqOzc3O&noteId=9i14rr7sNp)
> - More experiments with statistical analysis for the results: [kRiv Q3](https://openreview.net/forum?id=mJiPqOzc3O&noteId=mcVaWUSybA)
> - Comprehensive comparison with low fidelity data & smaller Re: [FhDJ Q4 and Q5](https://openreview.net/forum?id=mJiPqOzc3O&noteId=ZPmGLsqpnJ)
> - Stronger connections to the ScalarFlow experiment: [kRiv Q4](https://openreview.net/forum?id=mJiPqOzc3O&noteId=mcVaWUSybA)
>
> Overall, we present a principled, architecture-agnostic approach that enhances neural operators by explicitly incorporating fundamental physical knowledge into training, going beyond approaches that rely purely on data or large-scale pretraining. As reviewers FhDJ and kRiv noted, this is an important yet underexplored direction in the neural operator literature. Consistent with reviewer dWHm’s assessment that our method is well-motivated, simple but effective, we demonstrate substantial improvements in data efficiency, long-term stability, and out-of-distribution generalization across diverse PDE systems and architectures. Collectively, these results demonstrate that leveraging fundamental physics offers a powerful and previously untapped inductive bias for advancing scientific machine learning.

---

### Meta-Review · Area_Chair_2BvJ · 2026-01-09

**Summary:**

Reviewers generally indicate that:
1. The problem being studied is well-motivated and important to scientific machine learning.
2. Numerical experiments are have sufficiently good baselines and control well for the total cost of each method.
3. The statistical significance of the results is not presented rigorously enough.
4. The method of decomposing the PDEs is heuristic and it's unclear what constitutes "fundamental physical knowledge."

**Reviewer Concerns:**

The authors have addressed the majority of reviewer concerns regarding baselines and statistical significance. The authors have also introduced a metric to quantify "physical significance." While I think this is helpful, I stand with the reviewers concerns that it's largely heuristic. Nevertheless, this is an open problem in the field that is unlikely to be resolved in any one particular way. The present paper does make a significant contribution that can be useful to partitioners in this area.

**Reviewer Scores:**

Reviewer dWHm already raised their score to a 6, and I believe reviewer kris would have as well.

---

### Decision · Program_Chairs · 2026-01-26

Accept (Poster)